

# Comprehensive multi-hazard risk assessment in data-scarce regions. A study focused on Burundi

Jess Delves[1][2], Kathrin Renner[1], Piero Campalani[1], Jesica Piñon[3], Stefan Schneiderbauer[1][2], Stefan Steger[1], Mateo Moreno[1], Maria Belen Benito Oterino[4], Eduardo Perez[5] and Massimiliano Pittore[1].

[1]Eurac Research, Bolzano, 39100, Italy
[2]GLOMOS-UNU-EHS, Bolzano, 39100, Italy
[3]METEOSIM S.L., Barcelona, 08028, Spain
[4]Seismic Engineering Group (GIIS) UPM, Madrid, 28031, Spain
[5]IDOM Consulting, Engineering, Architecture, S.A.U., Bilbao, 48015 Spain

*Correspondence to*: Kathrin Renner (kathrin.renner@eurac.edu)

**Abstract**. The increased occurrence of multiple cascading and compounding hazards underlines the importance of integrated- and multi-hazard-based assessment approaches for the development of thorough strategies towards disaster resilience. To this purpose, a national-scale multi-hazard risk assessment was conducted between September 2020 and December 2021 for Burundi, focusing on the natural hazards flooding, torrential rains, landslides, earthquakes, and strong winds. This integrated multi-hazard assessment resulted in comparable nationwide provincial and commune-scale Annual Average Loss (AAL) values, further aggregated to provide a preliminary estimate of the resulting overall risk. Historical climatology (1990-2019) was computed, and a preliminary evaluation of the potential effects of climate change in the future period (2020-2049) was carried out. Data availability and reliability were challenging throughout the whole assessment and were tackled by integrating local authoritative sources with international and global resources. An up-to-date exposure model was implemented and complemented by an indicator-based socioeconomic vulnerability assessment. Furthermore, a data-driven statistical susceptibility model for shallow landslides has been derived at national scale. The consequent multi-hazard risk assessment provides an approximate picture of the expected nationwide risk distribution in economic terms. The results should support the identification of priority areas and actions for disaster risk management.





## 1 Introduction

'Understanding disaster risk' is the first of the four 'priorities of action' of the Sendai Framework for Disaster Risk Reduction 2015-2030. This priority emphasises that "policies and practices for disaster risk management should be based on an understanding of disaster risk in all its dimensions of vulnerability, capacity, exposure of persons and assets, hazard

characteristics and the environment." (UNISDR, 2015). Hazard risk assessments are a crucial component leading to an understanding of disaster risk and generating knowledge required for disaster risk reduction

(European Commission. Joint Research Centre., 2021). However, the increased occurrence of multiple cascading and compounding hazards underlines the importance of integrated and multi-hazard-based assessment approaches for the development of thorough strategies towards disaster resilience (Choi et al., 2021; Schneiderbauer et al., 2017; Zebisch et al.,

35   2023).

The activities presented here were funded by the International Organization for Migration (IOM) in the context of a larger European Union development program focused on Disaster Risk Reduction (DRR), with the aim of conducting a multi-hazard assessment and risk mapping of all 18 provinces and 119 communes of Burundi, with a more refined focus on five particularly vulnerable localities in the country. The five hazards prioritized by IOM and local authorities were flooding,

torrential rains, landslides, earthquakes and strong winds. The results of the assessment were aimed at decision makers, civil protection authorities and other stakeholders at national and sub-national levels to support planning, decision-making and prioritisation of Disaster Risk Management (DRM) investments and activities.

The multi-hazard risk assessment was conducted between September 2020 and December 2021. It faced multiple challenges and disruptions related to the COVID-19 pandemic, including travel restrictions and staff illness. A significant limitation was

the lack of availability of observed data and the lack of reliability of some datasets. Concerns regarding the reliability of datasets stemmed from discrepancies between similar datasets collected from different international sources. These data and research gaps necessitated an assessment approach based on proxies with the aim of providing an approximate nationwide disaster impact distribution, which was applied for torrential rains, strong winds and landslides. This integrated multi-hazard assessment resulted in comparable nationwide provincial and commune-scale Annual Average Loss (AAL) values for the

five hazards under consideration.

To characterise the current climatology and explore the possible effects of climate change in a near future, both a historical climate (1990-2019) and climate change modelling (considering future period between 2020-2049) were completed. The latter focused on precipitation and winds and included three different climate change simulations performed at 3 km spatial resolution for Burundi considering a business-as-usual RCP8.5 (Representative Concentration Pathways) emission scenario.

This paper presents the methodologies applied in conducting a national, integrated multi-hazard risk assessment in a data scarce environment and aims at supporting Disaster Risk Management (DRM) actors (e.g. national and provincial government, IOM) in prioritising actions and fundings. However, this proxies-based approach should be considered as an initial status reference for the country which should be progressively improved as more technological and DRM capacity





becomes available in the country and better disaster impact records are collected. The description of implemented activities
and results can support scientists and practitioners in identifying efficient approaches in similarly data-scarce environments.
In the next sections a brief introduction to the case study is provided, followed by the methodological descriptions. Two
subsequent sections present and discuss the most relevant results. Before summarising the conclusions and a potential
outlook, an additional section describes the activities related to climate change.

## 2 Study area

Burundi is one of the smallest and most densely populated countries in central-eastern Africa. Of its 27,834 km2 about 2,000
km2 are occupied by Lake Tanganjika, in the west of the country. It shares borders with Rwanda, Tanzania and the
Democratic Republic of Congo. It is divided administratively into 18 provinces and 119 communes which are in turn divided
into collines (literally, hills, in French). The economic capital is Bujumbura in the west of the country and political capital
Gitega in its centre. The country shows a varied relief with altitudes that range from 774 masl at the northern border of Lake
Tanganjika to 2,670 masl of the highest peaks of the Congo-Nile Mountain chain that divides the country north-south. The
country is subdivided in five eco-climatic regions (from west to east): the lowlands of the Imbo plain bordering the Tangan
which corresponds to an extension basin of the western Rift Valley; the rugged region of Mumirwa; the mountainous range
(Congo-Nile Mountain chain); the central plains and the Kumoso; and Bugesera lowlands (Fig. 1). Figure 1 also shows the
hotspot areas for which more detailed risk assessments for some of the hazards were conducted.




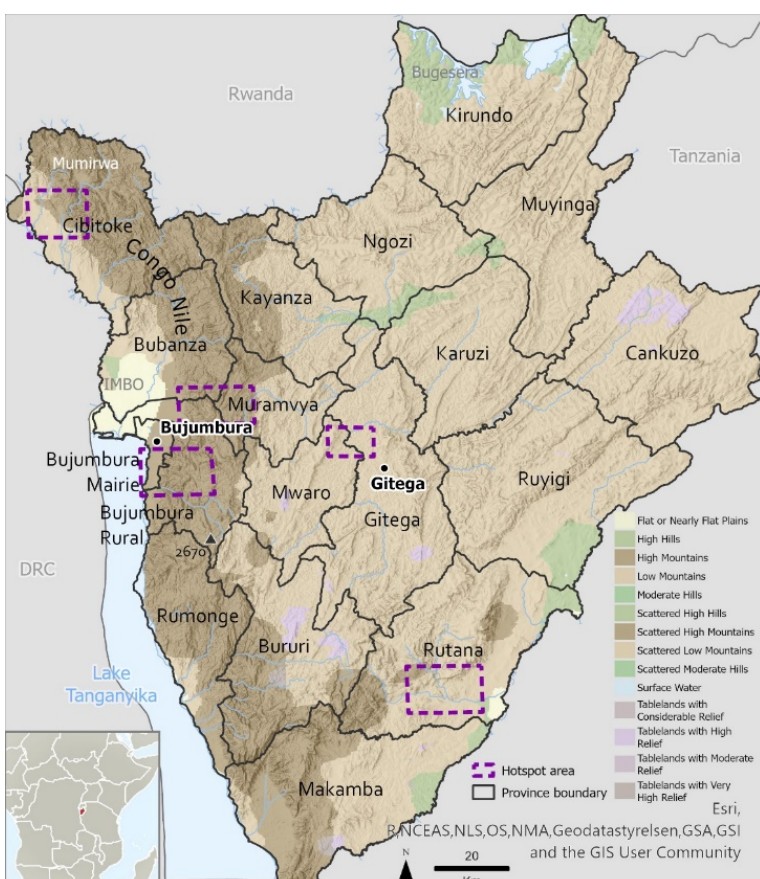

**Figure 1. Map showing the study area, its topography, administrative divisions at province level and the target/hotspot areas used in the sub-national hazard analysis.**

The economy of Burundi relies heavily on the agricultural sector, where, despite the scarcity of arable land, farming is practiced by 80% of the population, mostly for subsistence. Burundi's farming system is characterised by small farm sizes, with average lots of 0.5 hectares and a high land use intensity compared to other African countries (Dixon et al., 2019; Schneiderbauer et al., 2020). Nevertheless, agricultural productivity is low and food insecurity is high.

Amongst East African countries Burundi in 2010 had the lowest (13 percent) percentage of urban population, although

urbanisation is expected to increase (Dixon et al., 2019). Poverty is mainly rural and overwhelmingly affects small farmers, who lack technical skills, equipment, and financial capital. Approximately 1.7 million people were in need of humanitarian assistance in 2020 (OCHA, 2020). Investment in public services has not kept pace with rapid demographic growth, which is placing increasing pressure on resources and services such as healthcare and schools (ISTEEBU, 2017). Burundi in 2020 had a projected total population of around 11 million, a density of 442 inhabitants per km2 and a growth rate of 2.23 % in 2019

(ISTEEBU, 2013). Natural hazards and their impacts are the primary driver of human displacement in Burundi, with more than 80% caused by what is described by IOM as 'natural disasters' (IOM, 2020).



### 2.1 National DRR data context

According to the EM-DAT disaster data platform around 60 disaster events occurred between 1997 to 2020, 38 of which corresponded to the five hazards considered in this study (flooding, torrential rains, landslides, earthquakes and violent winds). The remaining events were associated predominantly with epidemic events and droughts. The reported number of deaths was collectively around 300 with approximately 230,000 affected people. Overall, torrential rains and floods contribute most to these figures followed by strong winds. Most of the meteorologically triggered hazards occur along the western provinces of the country, those that drain towards the Tanganyika Lake from north to south, although relevant disasters have been also reported across the eastern provinces of the country. The area is also subject to landslides, usually triggered by intense precipitation, as well as earthquakes.

To complement these data with locally updated records, IOM´s Displacement Tracking Matrix (DTM) platform began the systematic collection of disaster impact data in 2018. According to this database, during the 2018-2021 period around 9,500 houses were totally or partially destroyed and more than 40,000 houses were affected by different disasters. Additionally, around 130 casualties or missing people were reported, as well as around 215,000 affected and nearly 80,000 displaced people. The discrepancies between these two datasets (EM-DAT and DTM) are illustrative of the difficulty encountered by the study team in understanding the overall historic context of natural hazards and disaster impacts in Burundi.

### 3 Methodology

The assessment methodology is presented in Fig. 2. Since most of the considered hazards are highly influenced by climatic conditions, an analysis and modelling of the climatic driver in the historical period 1990-2019 was undertaken. A probabilistic approach was chosen to allow for the estimation of annual average loss figures for all hazards, with different return periods (RPs) selected to better capture the expected frequency/magnitude distribution of underlying processes. Whenever possible the individual risk components, namely hazard, exposure, and vulnerability, were explicitly considered and integrated. In the case of strong winds and intense precipitation a simpler approach based on empirically derived hazard/impact relationships was employed. For earthquakes and fluvial flooding in urban environments physical vulnerability models (fragility curves) from literature were used, while for the impact of fluvial flooding on agricultural areas and for landslides a simpler binary fragility model was used, due to a lack of consistent alternatives. National-scale models were employed in most cases, except for fluvial flooding, which was carried out in more detail in several hot-spots and later projected to other flood-prone areas in the country.



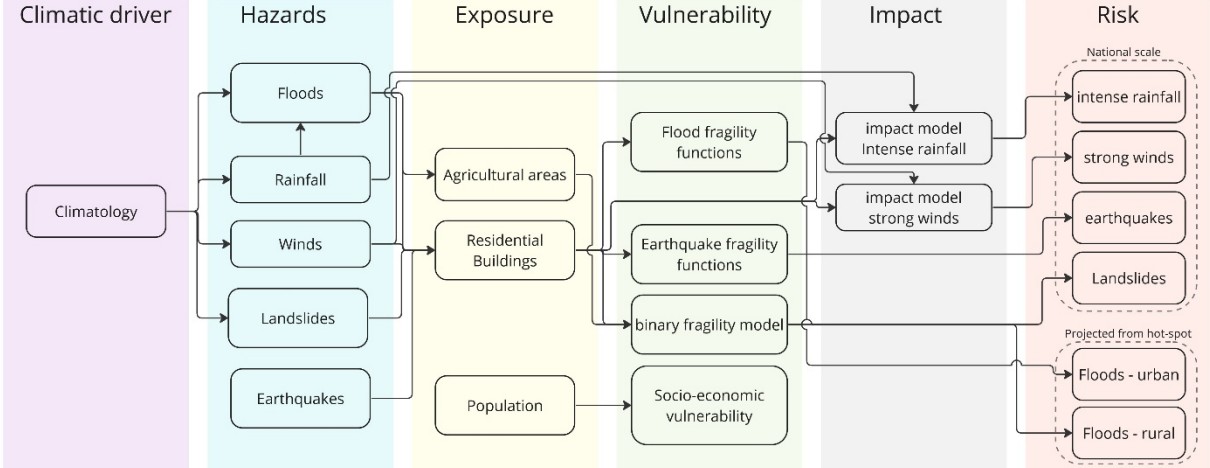

**Figure 2. Conceptual diagram of the methodology applied for the multi-hazard risk assessment.**

## 4 Climatic drivers

The climate of Burundi varies in accordance with the diverse altitude across the country. It can be described as humid tropical and is characterised by the alternation of a rainy period from October to May and a dry season between June and September. Precipitation increases with altitude, with the lowest rates approximately 500 mm along the Rusizi river and Imbo plain, with the maximum rainfall being approximately 2,200 mm in the highest regions. The average rainfall rate across the country is around 1,270 mm. The wettest period of the year usually takes place during April, with the largest number of rainy days (between 16-26). The yearly average temperature decreases with altitude, ranging from around 15.6°C in the high northern Kayanza province to 24.1°C on the Imbo plain. The maximum average monthly temperatures coincide with the end of the dry season (September-October), whereas the lowest average monthly temperatures occur during the dry season (MEEATU, 2012).

The study of climate in Burundi was based on a combination of large-area climate predictions and local downscaling techniques. Global Climate Models (GCMs) in the CMIP5 (Coupled Model Intercomparison Project 5), were adjusted to obtain more specific information at Burundi national and subnational scale, climate 'downscaling' was applied using a Regional Climate Model (RCM) called Weather Research and Forecasting (WRF) (Skamarock et al., 2008) in the version 3.9.1.1 (see section 10 for further details and a discussion on the future climate).

## 5 Hazard Analysis

5.1 Intense rainfalls and strong winds





The analysis of frequency and intensity of torrential rains and strong winds were assessed through the calculation of Intensity-Duration-Frequency (IDF) curves, which relate to the recurrency of a certain type of event (frequency), with a given intensity and duration.

The selection of extreme events intensity was implemented using the Peak-Over-Threshold (POT) method which considers all measurement above a chosen threshold. The 99th percentile of the time series was selected as threshold. The selected exceedances over the selected threshold are fit to a Generalised Pareto Distribution (GPD), which is the most appropriate distribution method for exceedances (Coles, 2001). Using POT increases the number of samples included in the analysis in comparison to other methods (such as Block Maxima) and reduces statistical uncertainties. The POT analysis provided the return levels for three return periods: 2, 10 and 50 years. It should be noted that the results show greater uncertainty the longer the return period. All the results have a spatial resolution of 3 km x 3 km according to the resolution of the models.

Torrential rains are intense precipitation events, usually from a convective nature (such as storms), that are associated with high risk of pluvial and river flooding as well as landslides. In the case of Burundi, the rainiest areas are those found at higher elevations. The fallen water flows from the mountains towards the Imbo plain. Historically, lowlands of the Imbo zone received the least rainfall per year. According to Nkunzimana et al. (2019) only 1.1% of daily precipitation from 1960 to 2010 registered in Bujumbura airport was greater than 30 mm, while 88 % of the daily precipitation was less than 5 mm.

Strong winds are classified according to the maximum speed the wind reaches in a specific time interval. They are usually associated with storms and often compounded with strong rains and can damage buildings and infrastructure. For the analysis of strong winds, the maximum wind gust was considered. The data supplied by the modelling includes the average sustained 10-minute wind speed. However, the WRF model does not directly provide information on the maximum wind gust, that is, the maximum wind speed in a time interval of 3 seconds for each time frequency considered. Therefore, to calculate the maximum gust for the country, it was approximated as the product of the average wind by a gust factor, $k_{gust}$ (Eq. 1):

$$v_{gust} = k_{gust} \cdot v_{10-minutal} \tag{1}$$

Where $k_{gust}$ has been set to 1.66 according to the recommendation of the World Meteorological Organization (WMO, see (Harper et al., 2010)).

**5.2 Pluvial flooding**

A national scale flood hazard map for Burundi was generated by combining historical and geomorphological approaches. The method consisted of three steps: (1) collection of historical information and data related to flood in the region; (2) preliminary analysis of flood frequency, magnitude and related impact from available information; (3) definition of possibly highly susceptible areas at national scale. Complementarily, (4) geomorphological analyses based on digital terrain model were carried out in those flood-prone areas that were insufficiently covered by the historical flood information.





Information on 64 flood events that occurred between 2000 and 2020 was collected and analysed. An exploratory data
analysis at province level indicated that the highest number of registered flood events were concentrated in the western
regions of the country, in the basins that drain to Tanganyika Lake. Almost 40% of the flood events were recorded in the
provinces Bujumbura Marie (19%) and Bujumbura Rural (20%) while the provinces of Bubanza (8%) and Cibitoke (10%)
also show a relatively high flood proneness. According to available information since the year 2000, flooding damaged
around 500,000 homes, directly affected 400,000 people and displaced 600,000 people (displaced people are not included in
the affected people group).

Percentile bootstrapping was applied to define high probability flooding areas associated to each flood event included in the
database. A preliminary map depicting the estimated flood hazard proneness areas was then developed.

Historic information related to spatially explicit data on rivers classified by their importance (i.e., hierarchy) and proximity
to populated areas were used to identify watercourses to be studied in further detail. Topographical stream analyses based on
a 10 m DTM allowed the extraction of streams with a hydraulic order higher than 3 for subsequent analyses. In addition to
all 3rd order streams within Burundi, smaller streams were also included in cases where the historical data depicted a past
flood event.

Finally, the flood proneness map was refined on specific locations that were not adequately covered by the historical
assessment due to lack of available information. Main activities entailed updating the land classification for the GIS
processing of watercourses and flood-prone areas (valleys and fluvial terraces), the geomorphological assessment of
riverbanks to identify erosive forms that may be indicative of historical flood events as well as the assessment of vegetation.
This latter feature is often related with historically flooded areas in combination with sedimentary areas along the riverbanks.
The resulting national-scale fluvial flood proneness map is shown in Fig. 3.



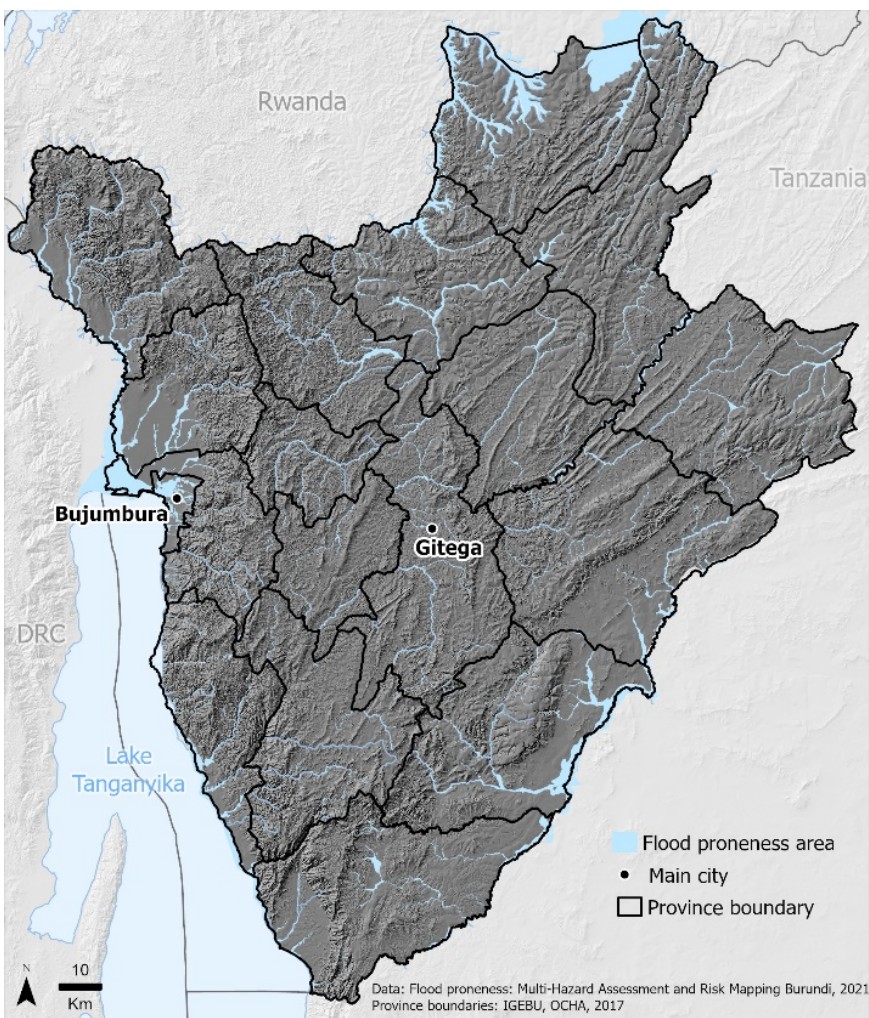

**Figure 3. National Flood Proneness Map.**

The flood proneness map was used to extend the economic risk calculated for five hotspot areas to the national scale. In order to calculate these local-scale risk hotspots a standard probabilistic flood-hazard assessment method was followed. This analysis comprises two processes, first a hydrologic study of the selected catchment area to estimate the recurrent (i.e. associated to the return period) water discharge rates that can be observed at a specific point along the river stream and, second, a hydraulic modelling activity to better characterise how the different recurrent discharge rates might generate flood events within the downstream river sector. The elaboration of the hydrological precipitation-runoff model was carried out using the Hec-HMS software developed at the Hydrologic Engineering Centre (HEC) of the US Army Corps of Engineers. The hydrological model was employed to estimate the recurrent precipitation for five selected catchment basins, for which IDF curves for each considered return period were developed. In the subsequent hydraulic study, the water levels and flow velocities reached during each rainfall event associated with a given return period were estimated. A set of inundation maps



was prepared for each hotspot area both for the current climate and the selected climate change scenarios. Figure 4 shows three of these maps for the 500-year return period (without considering the potential impact of climate change).

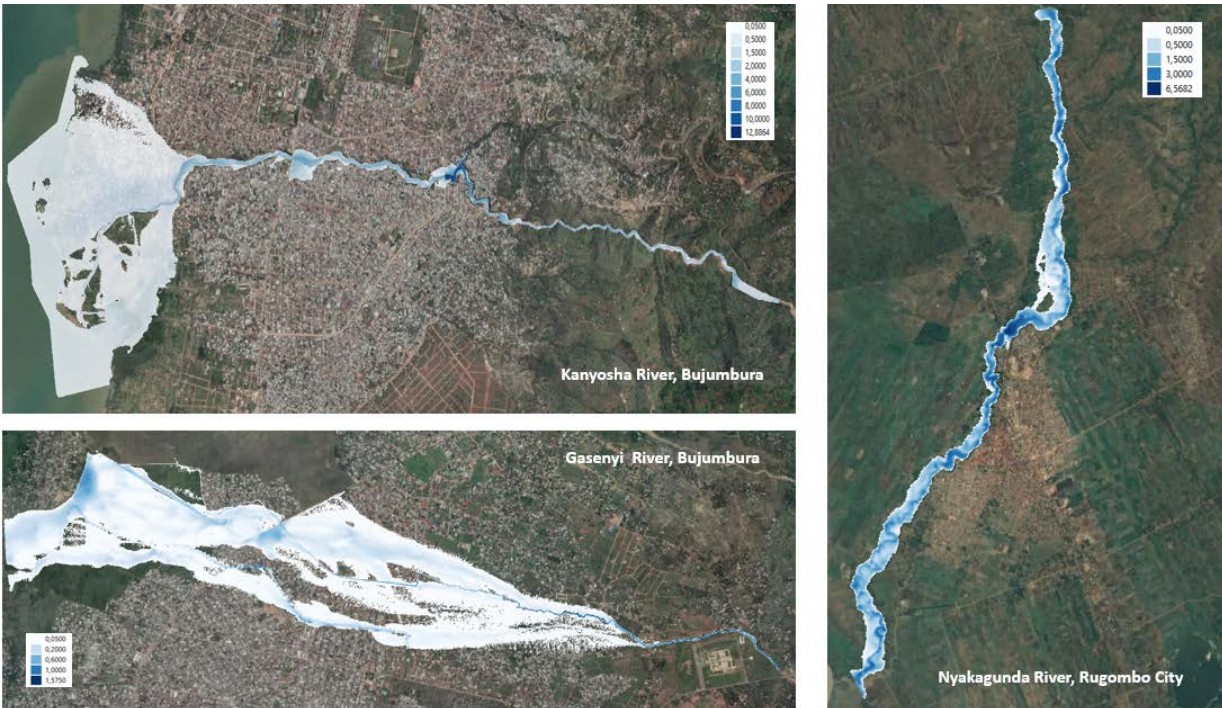

**Figure 4. Example of result of three of the five hotspots under study for 500-year return period scenario (© Google Earth).**

## 5.3 Landslides

The landslide hazard in Burundi was assessed in two phases: first generating a national-scale model of landslide susceptibility and then adding a time-recurrence and an intensity-frequency models.

### 5.3.1 Landslide susceptibility

Landslide susceptibility was addressed using a statistical, data-driven approach known as Generalised Additive Model (GAM). A GAM accounts for non-linear relationships between landslides and different environmental factors. The landslide inventory used for building the model consisted of landslide data partly based on studies conducted by Nibigira et al., (2013) including a subset of landslide information published by Broeckx et al. (2018) reaching a total of 770 landslides. The static environmental factors comprised a set of terrain derivatives (i.e. slope angle, topographic wetness index, slope aspect, curvature, relative slope position, geomorphons and terrain ruggedness) derived from the digital terrain model SRTM V4.3 and land cover maps. The resulting model was quantitatively validated using k-fold cross-validation and showed good performance in correctly identifying areas prone to landslides (median area under the receiver operating curve of 0.89 after 100 model runs).



In order to explicitly consider the dynamic role of precipitation as relevant landslide triggers in the area, the average of the maximum 3-day precipitation for every month was computed and used along with the landslide susceptibility to produce
landslide susceptibility maps as a function of the precipitation patterns throughout the year as shown in panels B and C in Fig. 5.

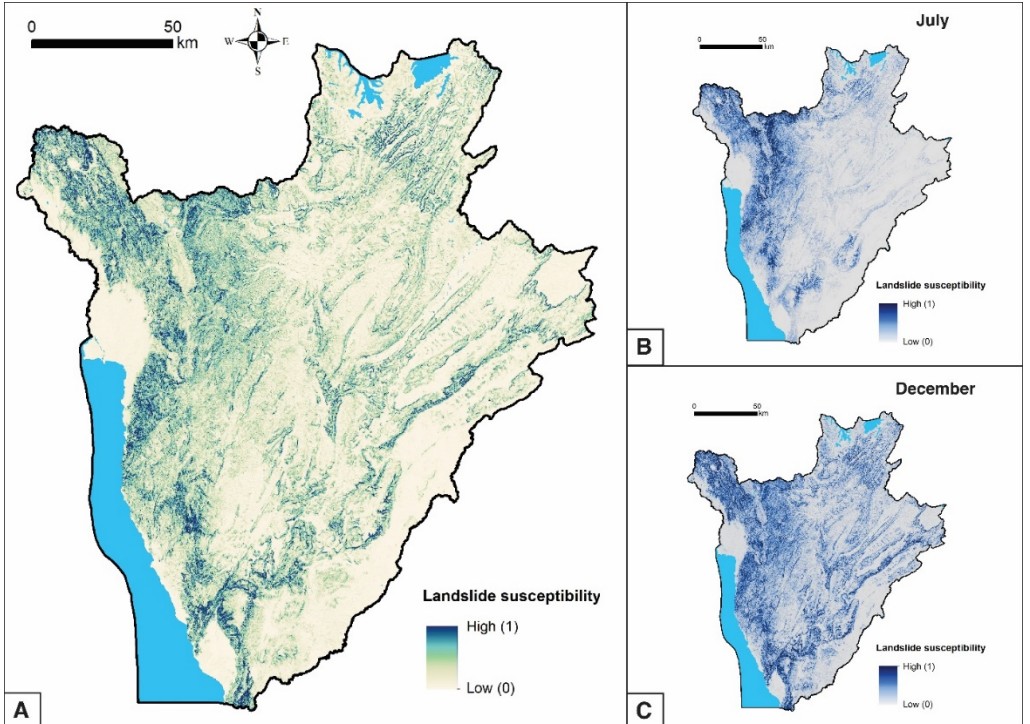

**Figure 5. National landslide susceptibility for Burundi (A) and landslide susceptibility for July (B) and December (C), months with respectively low and high precipitations.**

**5.3.2 Recurrence model**

A recurrence model was produced to estimate the expected frequency of the landsliding processes at national scale to assess the landslide hazard. Due to the lack of multi-temporal information on landside observations, a simplified recurrence model was developed using a heterogeneous Poisson Process model where the previously generated monthly landslide susceptibility drives the spatial occurrence of landslides. The model was calibrated by setting the average (expected) annual
number of events to 1,000 events.

Furthermore, it was necessary to characterise the distribution of event intensity, which in the case of shallow landslides can be approximated by their size in terms of spatial extent (Tanyaş et al., 2018). To achieve this, we used a power-law frequency-size distribution to identify the frequency of small landslide events with respect to larger ones. The parameters of this law were estimated by considering the existing inventory of past events, the most likely trigger (precipitation), the
235 literature Corominas and Moya (2008) and expert judgment. A fixed power exponent of -2.3 and a minimum event size of





100 m² were deemed representative for the subsequent risk analysis. The different landslide risk components (hazard, exposure, vulnerability) described in the respective sections were integrated into a Montecarlo numerical simulation procedure framework to derive a probabilistic assessment. A set of 1,000 simulation runs were generated for each month of a year. The output consisted of a set of (more than a billion) events, each with a geographical coordinate and a size assigned according to the underlying models previously described.

**5.4 Earthquakes**

The estimation of seismic hazard assessment followed the methodological approach known as Probabilistic Seismic Hazard Assessment (PSHA) integrating the results of source modelling, ground motion propagation and soil effects. The expected accelerations associated to the return periods of 100, 250, 475, 975 and 2,500 years (corresponding to exceedance probabilities of 40%, 18%, 10%, 5% and 2% in 50 years, respectively) were computed considering all known seismic sources within a 300 km radius zone of influence around Burundi. The probability of exceeding certain levels of ground motion (in terms of acceleration and spectral acceleration) due to all these sources was subsequently estimated. A logic tree was used to consider different attenuation models. This allowed the quantification of the effect of the epistemic uncertainty inherent to those models as well as the statistical variability associated to different calculation parameters.

The peak ground acceleration (PGA) and the spectral accelerations SA (T) for the aforementioned return periods across Burundi were obtained, computed for reference rock site conditions as well as considering local site effects. These maps do not represent the action of a specific earthquake, but the accelerations with the probabilities of exceedance mentioned by any future earthquake during the exposure time, which in this case was set at 50 years. Out of these results, the seismic response spectra to be used as seismic inputs for the five hotspot localities were obtained. The distribution of PGA, including local effects, for RP 475, which is considered the most representative return period for this hazard is presented in Fig. 6. The maximum projected PGA intensity is 0.37g. The seismic hazard maps including soil effects display spatial variations reflecting the lithologic diversity of Burundi. The areas with the highest seismic hazard are estimated in the alluvial deposits and sedimentary basins along the Imbo Plain, in the western part of Burundi, while the lowest expected hazard is estimated for the eastern part of the country due to lower seismicity ranges and the presence of harder soils.



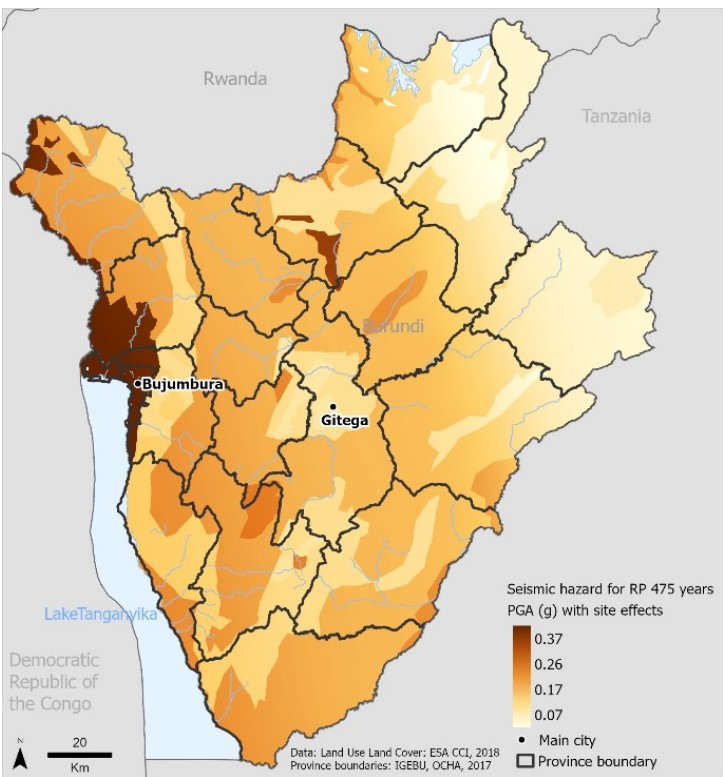

**Figure 6. Seismic hazard maps including site effects obtained for RP 475-years.**

## 6 Exposure

Exposure is a key component of risk, which is at the same time very challenging to assess and often insufficiently considered by practitioners (Pittore et al., 2017). Key exposure information considered in this study includes population, residential buildings and cultivated areas, and additional information on infrastructure was also collected (see (Campalani et al., 2023). The datasets used were either sourced from local authoritative sources or from trusted international projects and databases or derived through specific efforts by combining and processing existing datasets (as in the case of the population distribution and the characterisation of the residential building stock). Where more than one dataset was found, a comparative assessment was carried out and a recommended dataset was selected and shown on maps.

### 6.1 Population

Population in Burundi has grown from around 8 million people in 2008 (year of the last census) to around 12 million people in 2019 (projection of the National Statistics office). Thus, the population of Burundi can be considered highly dynamic. In order to provide a realistic representation of current population distribution, a 100 m resolution building constrained gridded population modelled dataset was chosen over the much coarser population numbers available for the 119 communes. Two





modelled population grids were analysed for quality against the available reference datasets. Based on the results of the accuracy assessment a calibration of the 100 m population modelled data was conducted. It is recommended to use the calibrated high resolution population grid sourced from Worldpop as a representation of population exposure. The maps in Fig. 7 show the population and building distribution.

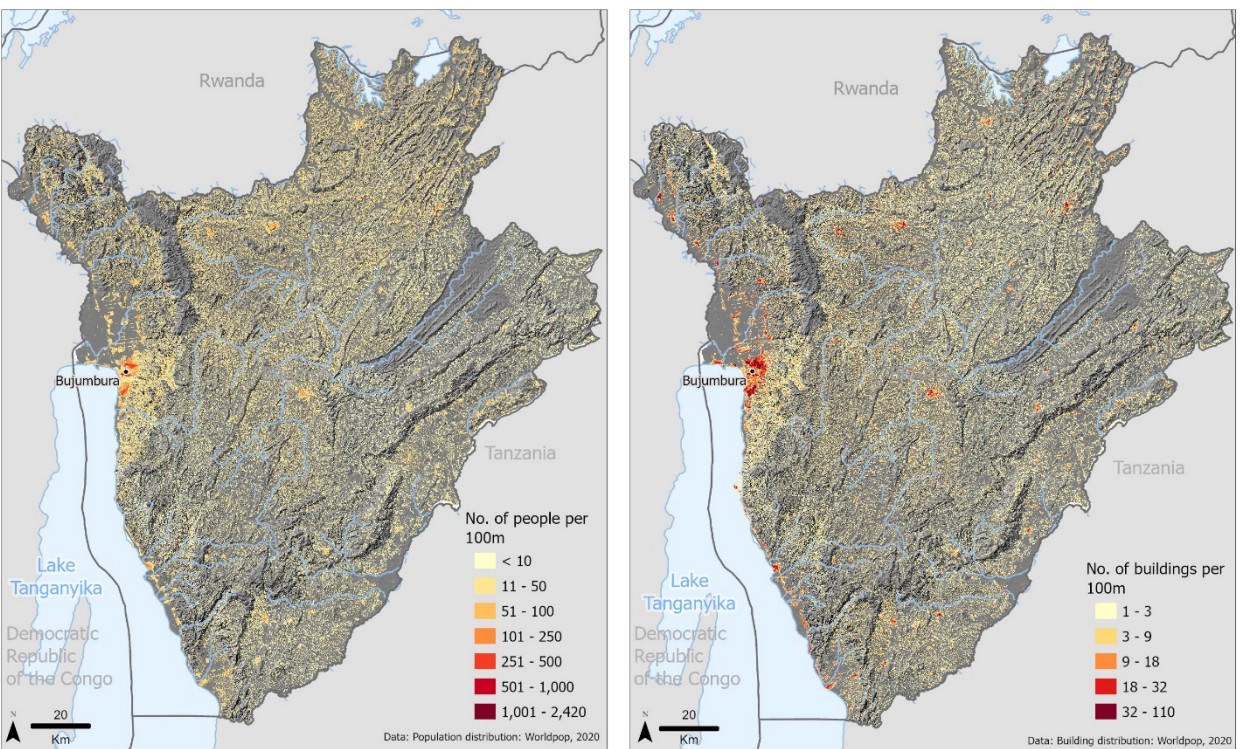

**Figure 7. Left: distribution of population. Right: distribution of residential buildings (number of buildings in a 100m x 100m square cell).**

### 6.2 Residential building stock

Given that no reliable nationwide building type distribution data was available from national institutions, an indirect projection approach was implemented. The process for establishing the number of buildings, their constructive taxonomies

and projected distribution at commune scale was implemented based on the following two information sources: 1) IOM's Displacement Tracking Matrix (DTM) colline-scale building screening survey dataset (2020) covering 602 collines and provided approximate construction type distribution estimates and building occupation rates; 2) Global Earthquake Model (GEM) Foundation[1] database which provided a province scale validated inventory of building taxonomies (Brzev et al., 2013) and; 3) field visits, during which a general building diversity scheme was set up based on material types and assumed

structure.

---

[1] https://maps.openquake.org/map/global-exposure-map/#3/21.04/10.99





The data was processed as follows: first, the GEM taxonomy construction-types were aggregated in accordance with the different vulnerability curves available for the different hazards focusing on a limited number of structural types. Secondly the DTM survey data were reviewed and aggregated by wall type. In total four wall type groups were defined: mud, bamboo/wood, adobe and brick/concrete. Lastly the GEM and DTM building types were merged resulting in the 10 baseline

taxonomies.

The number of buildings per commune was calculated by combining the occupation rates and the total number of buildings scaled by the population distribution. For validation the results were compared with projections for 2019 of the 2008 census-based GEM data resulting in a good match. Finally, the commune-scale building type distribution data was represented in pie-distribution charts as shown in for the provinces of Bujumbura Mairie and Cibitoke (see Fig. 8).


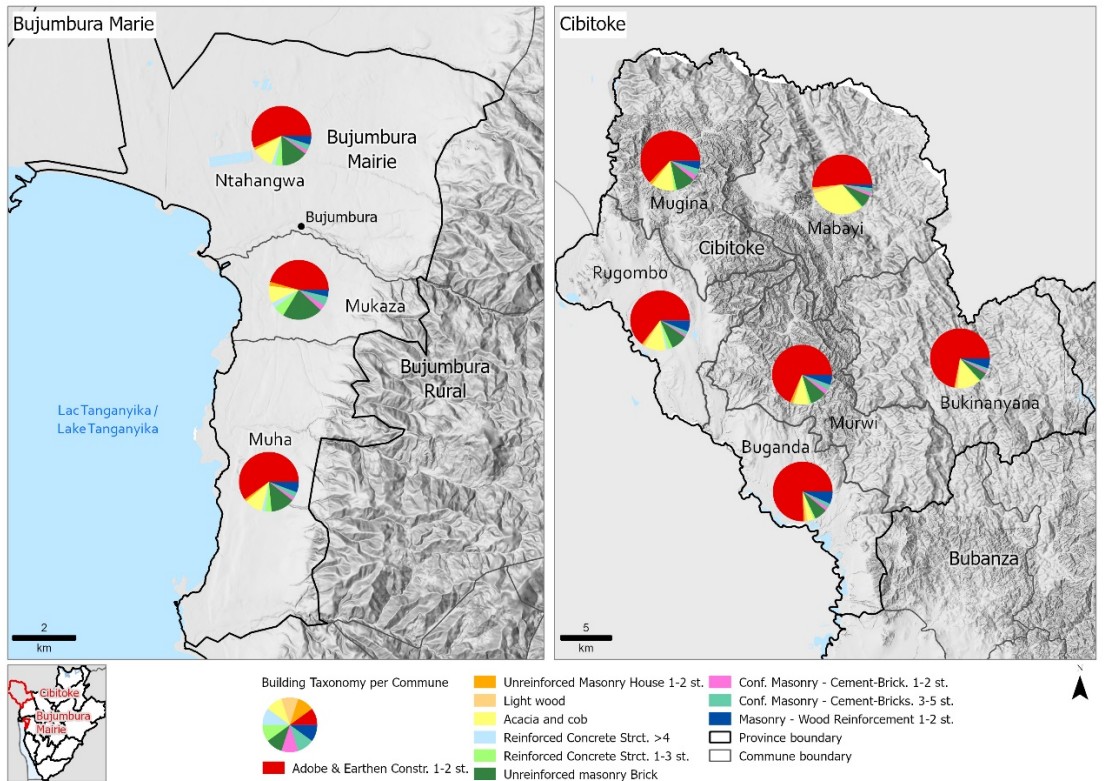

**Figure 8. Maps of commune-scale building taxonomies distribution maps of Bujumbura Marie (left) and Cibitoke (right) provinces (Data sources: administrative boundaries: ISTEEBU, 2017).**

## 6.3 Cultivated areas

With the largest proportion of Burundian people's livelihoods depending on income from agriculture, land used for farming, i.e. cropland, is an essential asset potentially exposed to natural hazards. Again, information on the distribution of agriculturally used land was not available from Burundian institutions. More than 60 % of Burundi is covered by cropland. Several land use classification products derived from very high-resolution satellites were analysed for accuracy. The mostly





visual accuracy assessment found the 100m landcover map for 2019 derived from Sentinel 2 satellite imagery best
represented land cover in Burundi. Pre-processing included the extraction of all the pixels classified as cropland and
presenting this information in a cropland distribution per colline map (Fig. 9).

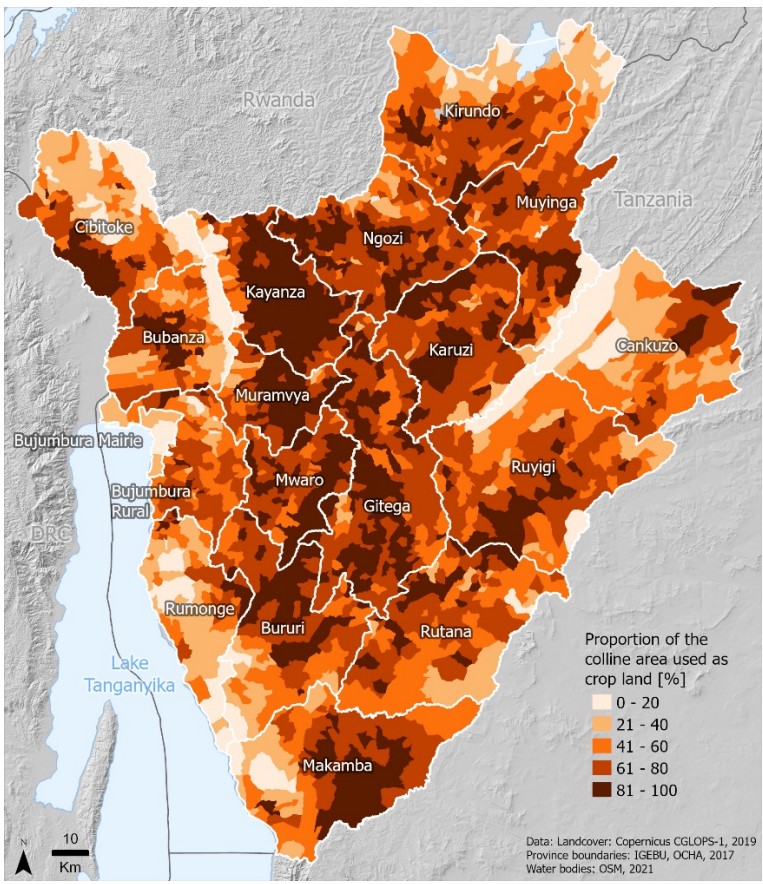

**Figure 9. Map showing the density of cropland per colline.**

## 7 Vulnerability

### 7.1 Physical vulnerability

Physical damage was considered only for residential buildings, considering their higher relevancy with respect to other
infrastructure in the scope of this risk assessment, and the general lack of data for assessment of other building types.
Different approaches, described in the following subsections, were adopted for the consideration of physical vulnerability,
depending on the specific hazard and considering only the impact on residential buildings.



### 7.1.1 Floods

Englhardt et al. (2019) propose vulnerability curves adapted to the African context (Fig. 10) which were used to estimate the relative damage factor associated to the water depth during fluvial flood events across Burundi. This vulnerability model has been employed in an Ethiopian context with building types comparable with those observed in Burundi.

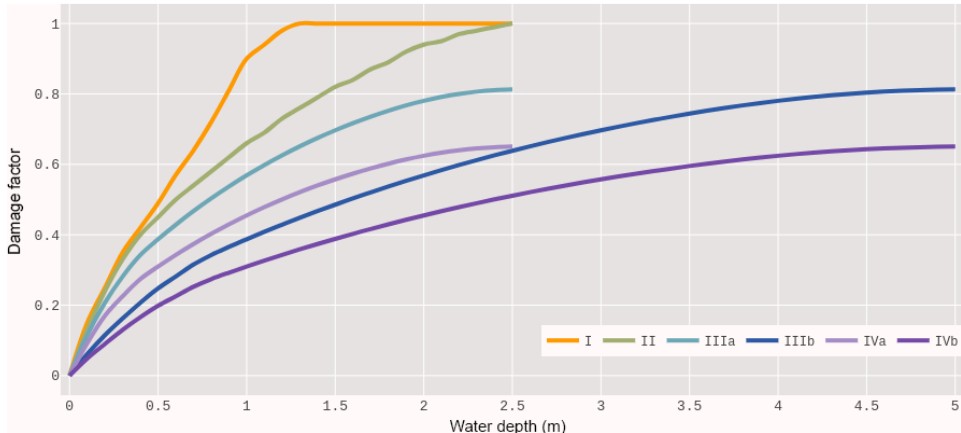

**Figure 10. Selected flood vulnerability curves for African contexts (Source: Englhardt et al., 2019).**

### 7.1.2 Earthquakes

Earthquake vulnerability was based on standard internationally validated fragility curves. To assign the capacity curves that represent the behaviour of the different construction typologies that were observed in Burundi with the best accuracy, the HAZUS scheme from Kircher et al. (2006) was considered as the most adequate.

### 7.1.3 Landslides

Considering the lack of fragility models compatible with the geographical setting and the use of a stochastic approach to estimate the risk, a simplified model for physical vulnerability (fragility) was adopted in the case of landslides. Residential buildings were assigned maximum damage (hence a loss equal to 100% of replacement cost) if their position fell within a given distance from the location assigned to a landslide event generated stochastically during the simulation, and zero damage (hence, a loss equal to 0) if outside. The given distance is equal to the radius of a circle whose area is proportional to the estimated landslide size.

### 7.2 Socioeconomic vulnerability

Burundi is characterised by generally high levels of vulnerability relative to other countries (INFORM, 2022). We understand vulnerability to be the degree of susceptibility of communities, systems or elements at risk and their capacity to cope under hazardous conditions (Birkmann et al., 2013). The assessment of socioeconomic vulnerability addresses



many intangible factors linked to characteristics of communities, such as knowledge and social network qualities. In this study, we also assessed coping capacity, understood as the ability of people, institutions, organizations, and systems, using available skills, values, beliefs, resources, and opportunities, to address, manage, and overcome adverse conditions in the short to medium term (IPCC, 2018). The lack of ability of systems to respond adequately to shocks and to evolve in
response to shocks (i.e. to be resilient) is assumed in this assessment to result in a vulnerability to all hazards. Therefore, socioeconomic vulnerability in this assessment is treated as hazard independent.

This socioeconomic vulnerability assessment (SEVA) used an indicator-based assessment framework applying existing data, supported with semi-structured interviews and focus groups with local experts. The latter qualitative methods served to identify context-specific factors of vulnerability and to validate and provide feedback on our assessment framework.

The assessment framework was adapted from the vulnerability and lack of coping capacity dimensions of the INFORM Risk Index developed by the Disaster Risk Management Knowledge Centre (DRMKC) of the European Commission (see (Marin Ferrer et al., 2017) for methodology). Adaptations were made in response to data availability and the project client's (IOM) requirements. The INFORM Risk Index is a composite indicator framework in which normalised vulnerability values are aggregated at each level of a hierarchical structure (through indicators, components, categories, dimensions). The INFORM
framework was chosen as it provides an accessible and transparent methodology, offers adaptability to user contexts and utilises existing data. Additionally, an INFORM subnational risk assessment in Burundi was published in 2020, of which some data were used in the SEVA.

The SEVA was conducted at two resolutions: colline and provincial. The respective spatial mapping uses (1) data from a survey commissioned by IOM Burundi in September-October 2020 in 532 collines (the DTM-DRR dataset) and (2) data
extracted from four national reports and from the INFORM 2020 Risk Assessment for the 18 provinces (EU-DRMKC, 2020). Proxies were used for indicators where data was not available.

Indicator data were normalised to a scale of 1 to 5 by assigning a vulnerability score to numbers (for metric data) and to statements/categories (for categorical data), with continuous class description from less vulnerable to more vulnerable (Fritzsche et al., 2014). To produce values for components of vulnerability, we used an unweighted arithmetic aggregation
(Fritzsche et al., 2014). At each step of aggregation, Min-Max normalisation transformed all values to scores ranging from 0 to 1 (Fritzsche et al., 2014). Values were not aggregated from the component to dimension level to avoid an over-simplification of results and potential misinterpretation by users. If more than 50% of input data for the construction of an indicator, subcomponent or component were missing, this was considered insufficiently complete and therefore not aggregated to the coarser levels. The results of the SEVA are presented as 11 maps showing the 11 vulnerability
components with both the colline and provincial assessment presented on each map. Every map is accompanied by a factsheet detailing the data sources, indicator calculation, and completeness of the underlying data.



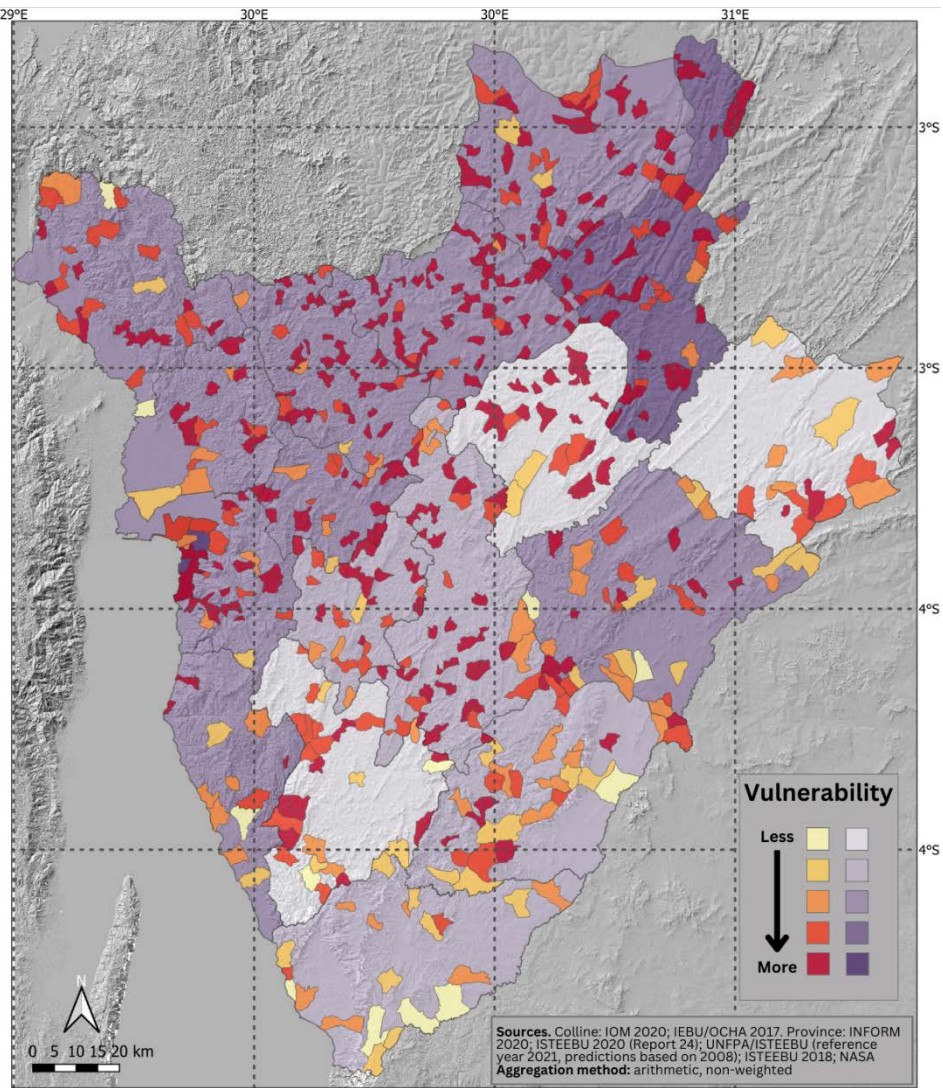

**Figure 11. Map showing data aggregated under the 'Population Characteristics' component of the SEVA. The underlying subcomponents are, for colline resolution population density and disadvantaged population and for provincial resolution population density increase 2050 and household composition.**

The results of the SEVA show a great heterogeneity in vulnerability both geographically and thematically over the two resolutions assessed, demonstrated by the frequency with which neighbouring collines show significantly different vulnerability levels across one indicator or subcomponent. This could be due to topographical and administrative boundaries which result in genuine differences in vulnerability and coping capacity between collines. The results suggest that differences in vulnerability factors are highly localised, but it must be stressed that these differences should be considered in the context of generally high and widespread vulnerability observed in Burundi. We recommend that users of the SEVA





results look into the underlying indicators and subcomponents when seeking to identify particular vulnerabilities or vulnerability hotspots. This is due to the loss of nuanced information caused by the aggregation of data to produce
components of vulnerability and the accompanying maps. This loss of information is more apparent the more subcomponents are aggregated.

Due to its limitations (see section 9), the results of the SEVA were not integrated into the monetary physical risk assessment. Consequently, the SEVA is to be considered as complementary to the overall risk assessment.

## 8 Results

In this section the results of the risk analysis for the individual hazards and their composition in terms of multi-hazard risk are provided and commented.

### 8.1 Multi-hazard map

In order to provide an intuitive description of the intensity and extent of the considered natural hazards, a series of national-scale, single-hazard maps were produced, each visualising the area exceeding the 85% percentile of the scalar intensity
distribution of the specific hazards (Fig. 12) on a 30m resolution grid. In the case of landslides, the hazard intensity is represented by the normalised susceptibility. These maps were then merged into a multi-hazard representation where each grid element is color-coded according to the number of hazards exceeding the 85% percentile, which ranges from zero to five. The resulting map provides a geographic representation of the potential hotspots for risk arising from multiple natural hazards, and is used to rank the individual communes, as shown in the table in Fig. 12 (listing the highest ranking 50
communes). This ranking does not consider the frequency/magnitude relationships of the individual hazards, nor their impacting mechanisms.





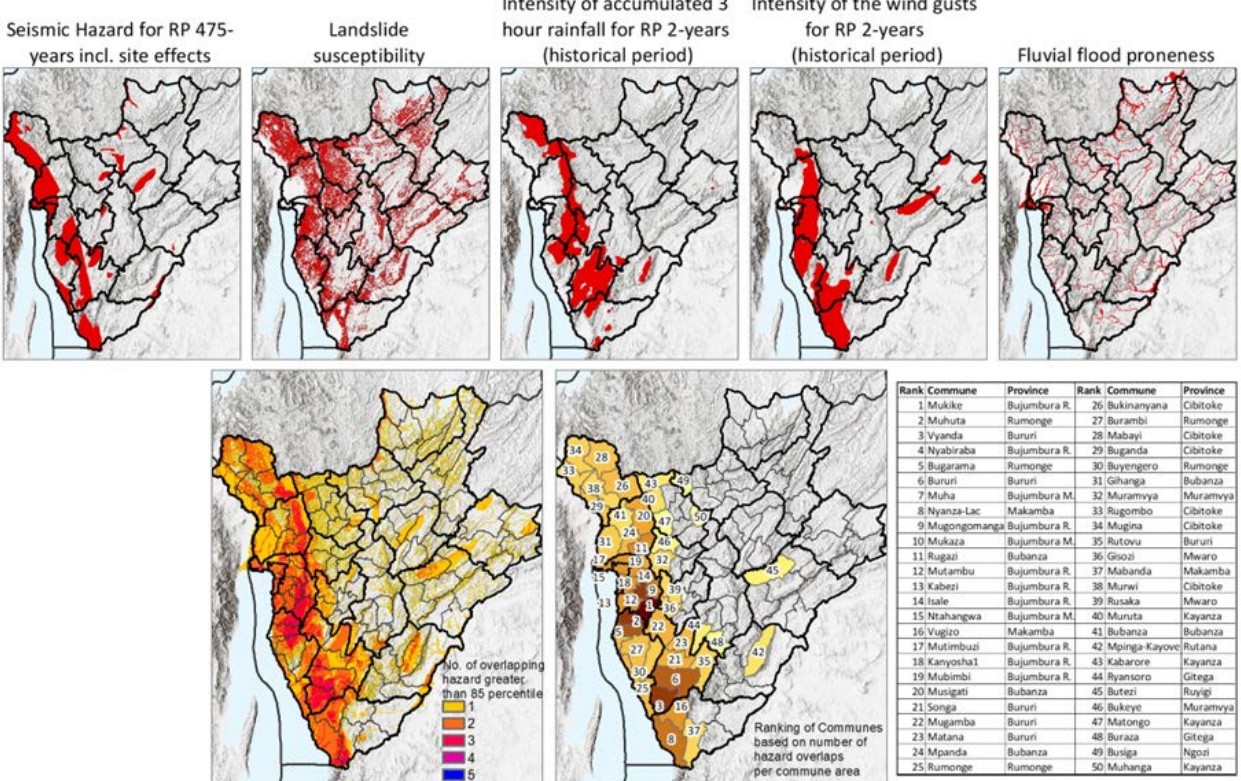

**Figure 12. Illustration of national-scale multi-hazard map generation. The top five maps show individual hazard maps exceeding the 85th percentile of the hazard distribution. The map on the bottom left shows the number of overlapping hazards and the map on the bottom right shows those communes potentially most prone to multiple hazards. The table shows a ranking of the 50 potentially most prone communes to multiple hazards (Data sources: administrative boundaries: ISTEEBU, 2017).**

To achieve a more quantitative comparison of the resulting risk, a probabilistic risk estimation was carried out for each of the five hazards at national scale (see Table 1 and Figure 13).





**Table 1. Overall national and provincial multi-hazard annual average loss (AAL) results.**

| | Provinces | Annual average loss (AAL) Projections (USD) | | | | | | Total MH AAL Projection (USD) |
|---|---|---|---|---|---|---|---|---|
| | | River Flood Urban Areas | River Floods Agricultural Plots (USD) | Torrential Rains | Strong Winds | Earthquakes | Landslides | |
| 1 | Bubanza | 406,080 | 529,651 | 188,237 | 186,655 | 1,370,944 | 107,165 | 2,788,733 |
| 2 | Bujumbura Mairie | 26,145,119 | 353,942 | 165,816 | 224,736 | 5,478,708 | 81,274 | 32,449,596 |
| 3 | Bujumbura Rural | 6,710,944 | 1,266,774 | 267,379 | 291,446 | 818,925 | 174,200 | 9,529,668 |
| 4 | Bururi | 0 | 222,139 | 193,957 | 146,354 | 479,563 | 101,514 | 1,143,526 |
| 5 | Cankuzo | 0 | 2,580,732 | 167,809 | 108,030 | 38,857 | 39,119 | 2,934,546 |
| 6 | Cibitoke | 73,440 | 743,557 | 265,203 | 204,222 | 782,860 | 152,959 | 2,222,241 |
| 7 | Gitega | 1,920 | 1,629,039 | 478,286 | 330,766 | 648,812 | 115,224 | 3,204,048 |
| 8 | Karuzi | 0 | 1,397,526 | 280,611 | 202,963 | 489,457 | 55,883 | 2,426,441 |
| 9 | Kayanza | 3,360 | 739,930 | 441,576 | 289,662 | 822,726 | 159,843 | 2,457,097 |
| 10 | Kirundo | 0 | 3,954,746 | 354,037 | 327,185 | 361,785 | 79,191 | 5,076,944 |
| 11 | Makamba | 72,960 | 2,258,549 | 210,448 | 206,543 | 578,559 | 106,169 | 3,433,228 |
| 12 | Muramvya | 14,400 | 333,015 | 183,732 | 128,696 | 440,721 | 95,867 | 1,196,431 |
| 13 | Muyinga | 0 | 2,872,453 | 373,780 | 328,027 | 127,523 | 117,477 | 3,819,261 |
| 14 | Mwaro | 0 | 539,516 | 187,278 | 125,984 | 339,128 | 63,279 | 1,255,185 |
| 15 | Ngozi | 26,400 | 6,752,309 | 522,261 | 333,025 | 500,122 | 124,148 | 8,258,265 |
| 16 | Rumonge | 31,200 | 261,622 | 205,306 | 213,485 | 1,295,037 | 111,243 | 2,117,892 |
| 17 | Rutana | 14,400 | 3,638,219 | 209,734 | 146,438 | 353,177 | 71,975 | 4,433,943 |
| 18 | Ruyigi | 111,611 | 2,984,089 | 274,786 | 192,791 | 380,410 | 63,084 | 4,006,770 |
| | TOTAL | 33,611,834 | 33,057,807 | 4,970,237 | 3,987,009 | 15,307,315 | 1,819,614 | 92,753,816 |
| | MH AAL TOTAL (USD) | 92,753,816 | | | | | | |
| | Burundi GDP 2022 (USD) | 3,340,000,000 | | | | | | |
| | AAL/GDP (%) | 2.78% | | | | | | |

To explicitly consider the contribution of exposure and vulnerability and obtain a consistent geographical representation of
expected multi-hazard risk, an aggregated spatial risk density index (in USD/km²) was computed for each commune by
dividing the aggregated AAL value by the geographic extension (km²), the results of which are shown in Fig. 13.





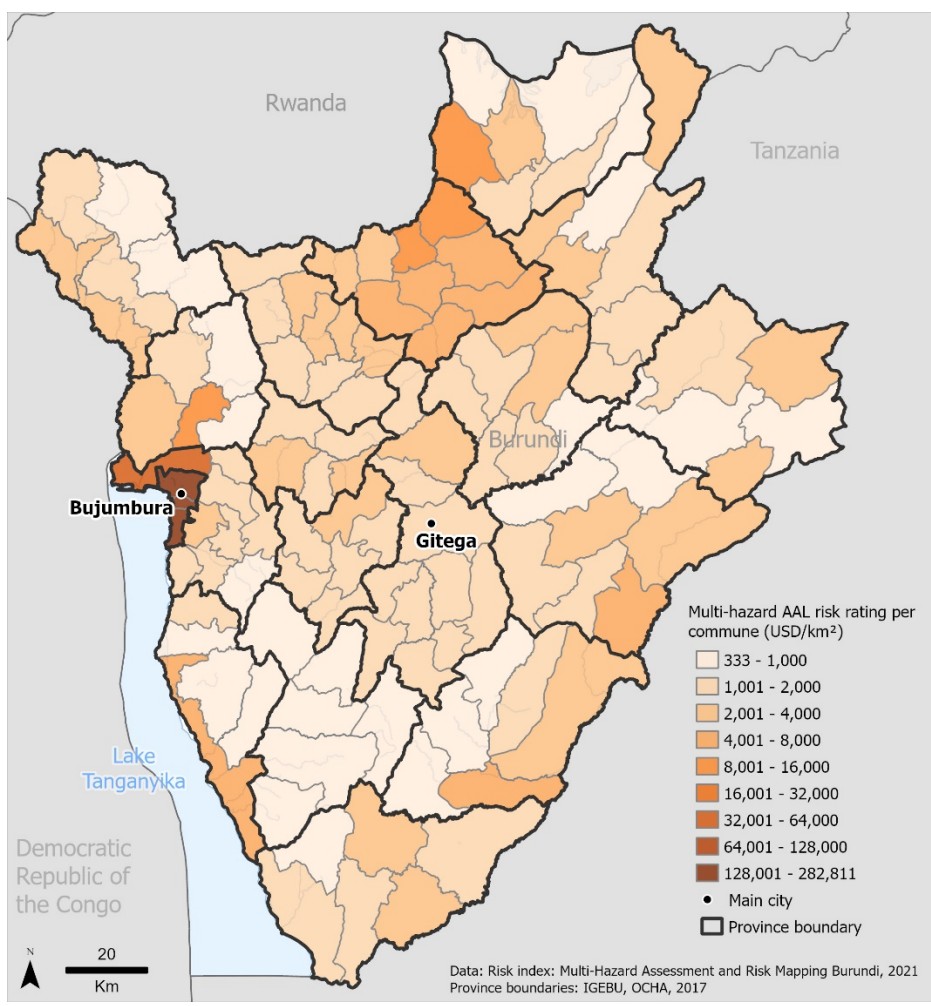

**Figure 13. Nationwide commune level multi-hazard AAL risk ratings with all hazards included.**

## 9 Discussion

The multi-hazard map (Fig. 12) highlights the communes with the larger number (in terms of their uppermost percentile) of overlapping natural hazards. This representation indicates a number of potential risk hotspots in the western part of the country, along the Congo-Nile Mountain range and bordering the Tanganika lake, where earthquakes, intense precipitations and strong wind gusts are possible. When a more comprehensive multi-hazard risk assessment is carried out, in terms of economic impact and also considering exposure and vulnerability, a different picture emerges (Fig. 13). The annual average

loss (AAL) is more evenly distributed across the country, with a pronounced hotspot in Bujumbura, as expected giving the concentration of assets, with other risk hotspots in the Ngozi and Kirundo provinces in the north, as well as in the eastern areas. The single-hazard AAL estimates are driven mostly by floods, impacting either urban areas or cultivated areas, and by earthquakes. The contributions of the Bujumbura provinces can be explained due to the concentration of population and





associated infrastructure in these provinces. With regards to Ngozi, the significant AAL of 6.5 million USD is mainly due to
potential losses to agricultural crops. Landslides contribute to multi-hazard risk comparatively less than other hazards, with
an estimated 0.005% of GDP. The highest single contribution (0.78% of GDP) is associated with fluvial floods in
Bujumbura Mairie. It should be noted that these assessments assume complete independence among the hazards, refer only
to direct physical impacts on residential buildings and agricultural plots, and do not account for climate change, and as such
should be considered as strong indications towards improved disaster risk management. Since natural hazards are expected
to strongly affect people and communities, a separate socio-economic vulnerability assessment (see 7.2 and Fig. 11) was
carried out based on the available data ranging from 2015-2020 and on consultations with local authorities and experts.

## 10 Climate Change

In order to understand the potential trends of risk in the future due to the evolution of climatic drivers, a study of climate
change in Burundi was carried out based on a combination of large-area climate predictions and local downscaling
techniques. Global Climate Models (GCMs) in the CMIP5 (Coupled Model Intercomparison Project 5) were used to obtain
information about how the climate could change at continental scale. In order to adjust such large-area climate prediction to
obtain more specific information at Burundi national and subnational scale, climate 'downscaling' was applied using a
Regional Climate Model (RCM) called Weather Research and Forecasting (WRF), version 3.9.1.1 (Skamarock et al., 2008).
By combining the broader predictions from CMIP5 with the precise details that the WRF model's downscaling method
provides, a more comprehensive picture of how climate change could impact the specific area of Burundi can be obtained.
The methodology is described in Fig. 14, and assumes as baseline the historical period 1990-2019, and a short-term future
period from 2020-2049.

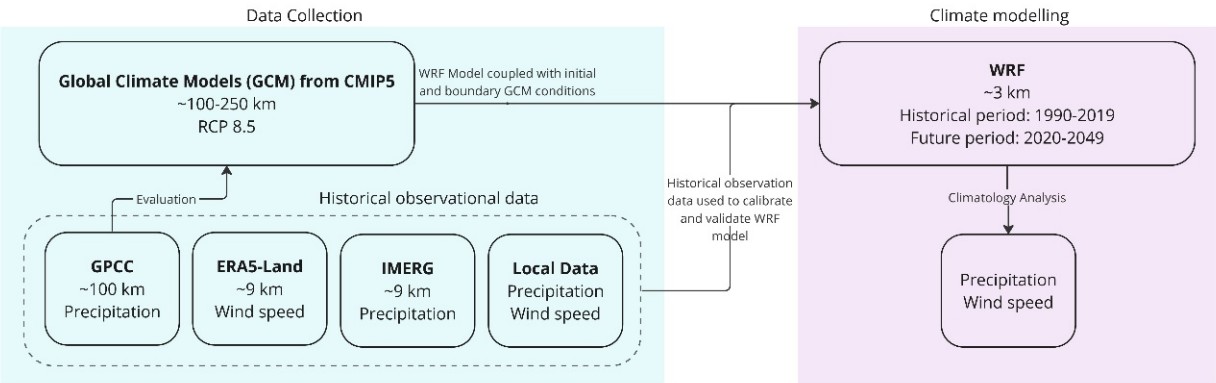

**Figure 14. Climate change dynamical downscaling methodology.**

Three distinct global models were downscaled using the WRF model, namely MPI-ESM-MR, CNRM_CM5, and CanESM2.
These three models were selected amongst the available global climate models to accurately reproduce the essential



characteristics of the current climatic conditions in Burundi's historical period, and to effectively capture the range of climate change signals present in the entire ensemble of GCMs.

As for the regional WRF model for Burundi, the model displayed a strong correlation in replicating the annual precipitation cycle, albeit with a tendency to overestimate this atmospheric variable during the rainy season. Similarly, the model exhibited good correlation with monthly averages of daily precipitation and the count of rainy days per month. Concerning wind speed, the model aligns well with scientific recommendations for mean daily wind speed and the 90th percentile of mean daily wind speed.

## 10.1 Impact of climate change on precipitation

Bujumbura Mairie is the province where the greatest increase in precipitation is expected with a projected relative increase in the 3-hourly precipitation of up to 58.1% for the return period of 50 years.

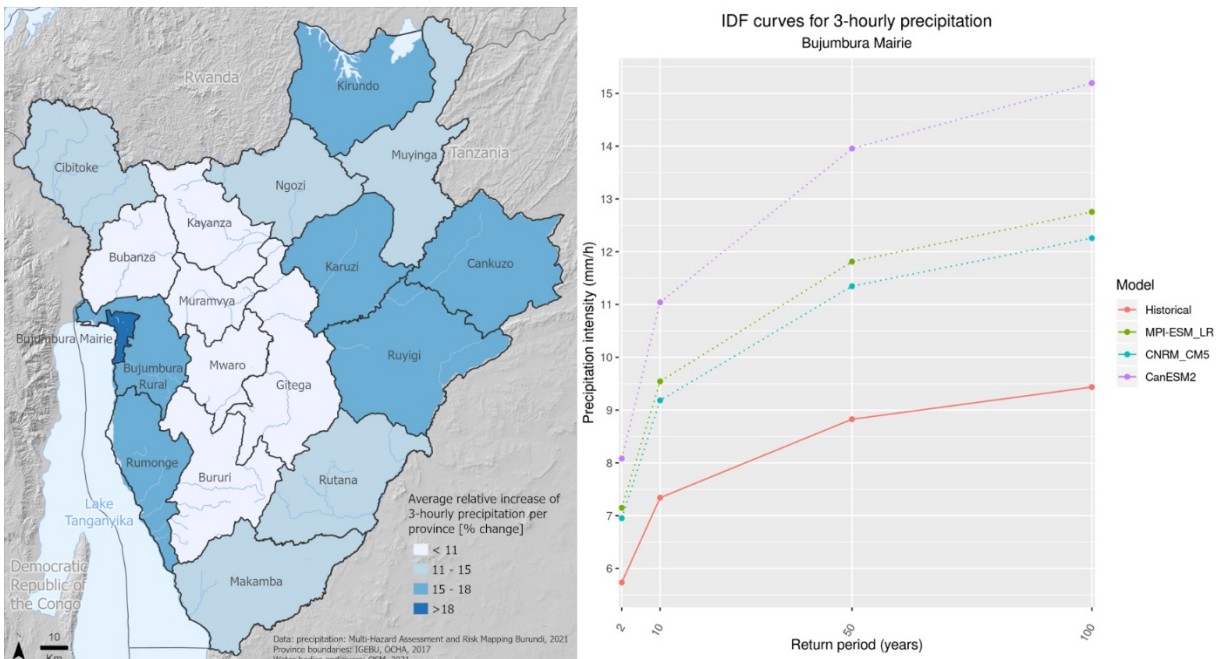

**Figure 15. Left: Provinces according to the projected increase of intensity and frequency of the 3-hourly precipitation. Right: Precipitation intensity for a duration of 3 hours predicted for the Bujumbura Mairie province by the forced WRF model with**
**different global and return periods.**

In Fig. 15, left, a summary of the results by province is shown, considering the average of the three future projections and the three return periods considered (2, 10 and 50 years). Darker blue colours indicate greater relative increases in the 3-hourly rains expected in the future compared with the historical value. The analyses indicate (e.g. Fig. 15 right) that for the same return period and duration, torrential rains in Burundi will be more intense (or alternatively, that precipitation of a given high
intensity will be more frequent). The smallest increases in the intensity are projected in the central part of the country (Kayanza, Mwaro, Muramvya, Bubanza, Gitega and Bururi).



## 10.2 Impact of climate change on wind

Figure 16 shows a summary of the results obtained by province, considering the average of the three future projections and the three return periods considered. Red indicates the relative increases in strong winds expected in the future when
compared with the historical value. Blue indicates the relative decrease in the intensity and frequency of strong winds in the future. Darker colours indicate greater decreases or increases. In the case of no conclusions, the provinces appear in white. In all the central provinces (Bururi, Cibitoke, Gitega, Karuzi, Kayaza, Makamba, Muramvya, Mwaro, Ngozi, Rutana and Ruyigi) the intensity of the wind gust is expected to increase when considering the same return period. The greatest increases in intensity are projected for Kayaza, Karuzi and Rutana provinces. For Bubanza and Bujumbura-Mairie provinces, small
decreases in the strong winds are expected. There is no agreement of the models for the northeast part of the country (Cankuzo, Muyinga and Kirundo) and the west provinces of Bujumbura-Rural and Rumonge for strong winds.

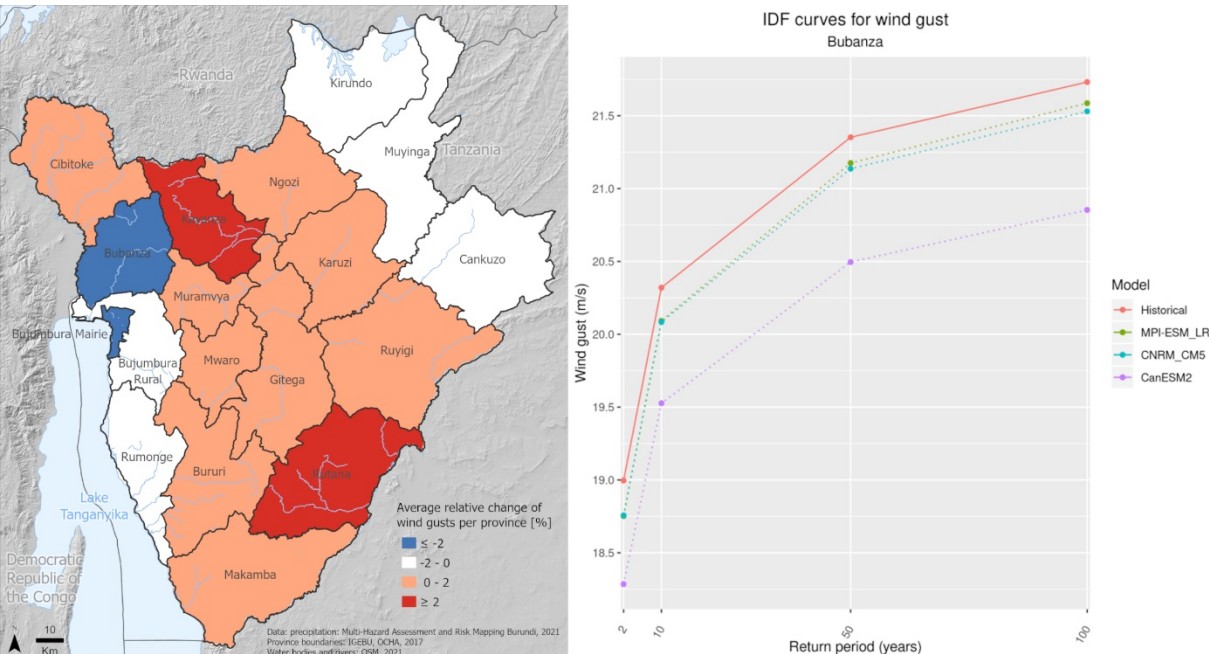

**Figure 16. Left: Provinces according to the projected increase (red) or decrease (blue) of intensity and frequency of the wind gust. Right: Wind gust predicted by the forced WRF model with different models for the Bubanza province and return periods.**

## 11 Conclusions and outlook

In order to explore the impact of natural hazards in Burundi and their distribution at different geographical scales, two approaches were followed. Following a single-hazard assessment, a multi-hazard map highlighted the communes with higher numbers (in terms of their uppermost percentile of overlapping natural hazards. This representation indicates a number of potential risk hotspots in the western part of the country, along the Congo-Nile Mountain range and bordering the Tanganika
Lake, where earthquakes, intense precipitations and strong wind gusts are possible. A more comprehensive multi-hazard risk



assessment was carried out, in terms of economic impact and also considering exposure and vulnerability, a different picture emerged. The annual average loss (AAL), in fact, is more evenly distributed across the country, with a pronounced hotspot in Bujumbura, as expected giving the concentration of assets in the city, with other risk hotspots in the Ngozi and Kirundo provinces in the north, as well as in the eastern areas. An aggregated AAL of over 92 million USD was estimated, amounting

to around 2.5% of Burundi's 2022 annual GDP. The single-hazard AAL estimates are driven mostly by floods, impacting either urban areas or cultivated areas, and by earthquakes. These assessments assume complete independence among the hazards, refer only to direct physical impacts on residential buildings and agricultural plots, and do not account for climate change, and as such should be considered as strong indications towards improved disaster risk management. A separate socioeconomic vulnerability assessment was carried out based on the available data ranging from 2015-2020 and on

consultations with local authorities and experts and provides complementary information on the ways natural hazards are expected to affect exposed people and communities. Since socioeconomic vulnerability is a dynamic process that evolves over time in response to changing local conditions, the monitoring of these dynamic processes is better able to represent the true vulnerability of a community. It is therefore suggested that the SEVA be updated as new data becomes available and new indicators added.

In general, data scarcity was a significant hurdle to overcome in the multi-hazard assessment of risk; several desired datasets were not accessible, did not exist or were incomplete, and were therefore not included. Data collection and field visits were not possible due to travel restrictions related to the COVID-19 pandemic as well as COVID-19 diagnoses within the team delaying (or leading to the cancellation of) fieldwork. To mitigate this, it was attempted to validate the conceptual approach and data sources with local experts, but this was limited due to the difficulty in engaging relevant informants. The use of

large-area and global datasets, most from authoritative sources, proved useful to set the stage for more in-depth assessments where possible, or to achieve a baseline result otherwise.

A preliminary evaluation of the potential impact of climate change has been carried out based on a combination of large-area climate predictions and local downscaling techniques and focusing on expected trends in precipitation and wind in the short-term future period from 2020-2049, considering as baseline the historical period 1990-2019. The mean daily precipitation

projections show a robust increase in the mean daily precipitation, ranging from 0.2 to 3 mm, compared to the historical period (1990-2019). The highest relative changes are found in the Imbo Plane, ranging from 25 to 40%. This trend is projected mainly for the rainy season. The eastern part of the country where the dry eastern plateaus are located also show high relative changes in the mean daily precipitation values. Regarding the total contribution from wet, very wet, and extremely wet days to the total precipitation, for several areas, such as Bujumbura Mairie, the north of Bujumbura Rural and

the south and central areas of Bubanza, as well as some areas of Kirundo and the north part of Ruyigi, the models show a positive trend in the expected changes with increases between 6-14% in the percentage of precipitation accumulated during the wet and very wet days. The contribution of the extremely wet days varies from 4 to 10%. Results from daily wind speed



during gusty days (days with wind speed values above 90th percentile) show a slightly decreasing trend, but in general, the wind speed values of the gusty days are expected to be comparable with the historical period, with moderate winds reached
during the windiest days.

Notwithstanding the limitations implicit in the narrower scope of the economic perspective on risk, the approach based on AAL allows for a consistent aggregation of risk arising from natural hazards with very different characteristics (e.g., intensity and frequency/magnitude distributions). This approach, being probabilistic in nature, also allows for the broad consideration of the aleatoric uncertainty underlying the considered processes. On the other hand, epistemic uncertainty is
significant due to the limited capacity in the modelling of some of the hazards and to the lack of context-specific data. In particular the potential effects of climate on future risk conditions should be directly accounted for in the AAL estimates. Also, the assumption of independence of hazards should be subject to further studies in order to better account for cascading and compounding mechanisms in the impacts, e.g. the triggering of landslides due to intense precipitation and earthquakes, and the high probability of compounding across the impacts related to hydrometeorological hazards.

*Competing interests.* The contact author has declared that none of the authors has any competing interests.

*Acknowledgements.* The activities described in this paper were funded by the International Organization of Migration (IOM), Bujumbura, Burundi. The authors would like to warmly thank, Sandrine Dhenain, Blandine Arvis, Mathieu Vinet, Lydia Rincón de la Rosa, Fanny Overlack, Alexis Nikiza, Léonidas Nduwayo, Celeus Rwishinza, for the continuous help
throughout the project. A particular acknowledgement to Carlos Villacis Villafuerte and Emmanuel Noel for the constructive support.

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
