# Peer review of "Comprehensive multi-hazard risk assessment in data-scarce regions. A study focused on Burundi"

_EGUsphere, 2024_

## Referee Comment (RC1)

The research by Deves and co-authors on an assessment of muti-hazard risks is welcome for a country like Burundi. Burundi is a small country of tropical Africa whose population is particularly exposed and vulnerable to natural hazards; these concerns being on an increase due to the rapid growth of the population. One of the reasons of this exposure is a lack of information on the presence of these hazards. Providing maps that locate these hazards comes therefore as a handy for the stakeholders.

Burundi, like many other regions of the Global South is clearly data-scarce. This scarcity of information is not only reflected on the lack o f observations on the hazards (inventory, databases, etc), monitoring (climate, seismicity), but is also due to a lack of ancillary information (e.g. lithology, soils, etc.). In such a case like Burundi, insisting on the data-scarcity context for the hazard risk assessment is therefore a very valid point made by the co-authors of this research.

In this research, the natural hazards that are studied are diverse: flooding, torrential rains, landslides, earthquakes, and strong winds. In addition, a preliminary assessment of the potential effects of climate change is carried out.  A specific focus is also put on shallow landslides for which a national-scale data-driven model has been derived.

Overall, this research falls very well within the scope of NHESS. However, when going through it, I have pointed some issues that should be addressed in, I think, a better way. They are listed below. Note that as an expert working on geo-hydrological hazards (landslides, flash floods, gully erosion) in the region, my comments are more specifically oriented towards these processes. For the other hazards, my critical look is definitely less developed.

- **The data-scarcity context and the lack of use of available data and knowledge in the assessments**

Although we are in a data scarce context, there are some available datasets  and knowledge that, if not used, the authors should refer to and discuss.

For example, Depicker et al (2020) produced a comprehensive landslide susceptibility map for a region that covers a large part of Burundi. This assessment is based on thousands of landside observations and has proven to deliver excellent predictive performances. The model is available online free access here: https://zenodo.org/records/5027004

Besides that, landslide inventories available online can be found here:

- Shallow landslide inventory compiled by Depicker et al. (2021a) , with more than 800 entries for Burundi
- Extra landslide inventory on a specific landslide event (Deijns et al., 2022): https://zenodo.org/records/7198322

For the region of Bujumbura in particular, comprehensive efforts on mapping landside processes have been done with the location of more than 1000 features (Kubwimana et al., 2021).

For the same region, floods have been documented and analysed in details by Nsabimana et al. (2023).

The relationship between landslides and flash floods has been investigated, either as cascading or compounding processes (Nibigira et al., 2018; Kubwimana et al., 2021; Deijns et al., 2021; 2024).

For earthquake hazard, reference the work of Delvaux et al (2017) on seismic hazard assessment for a region that includes the whole Burundi is also missed.

Vulnerability and risk to hazards have also been studied, either from a local perspective (e.g. Nsabimana et al, 2023) or from a regional one (Depicker et al., 2021b).

The examples I provide here a not an exhaustive list for all the hazards. However, what I want to point is that in its current version, this research misses the point of being framed around the state of the art knowledge in the region. Besides the implication it has for the science it itself, it is also problematic for the confusing message it could bring to the stakeholders. Why should new assessments be made if they are not discussed/compared with respect to current ones? Why should the local knowledge and expertise from researchers based in local institutions, even when available online, not be used?

- **Robustness of some analyses and lack of methodological information**

Once again, I am not able to put a critical eye on all the types of hazard assessment. However, when it comes to landside assessment for which I have more knowledge I have some concern. The analysis is based on a data-driven approach that is calibrated from an inventory of 770 landslides. The authors say that this information is from Nibigira et al (2013 – a non peer-reviewed information that cannot be accessed) and from Broeckx et al (2018). The PhD thesis of Nibigira (2019 - https://orbi.uliege.be/profile?uid=p125344 ) shows that he has mapped a total of 94 + 338 = 432 landslides over two well constrained regions of Burundi (see page 66 of the thesis). The data by Broeckx et al (2018) provide 204 entries for Burundi. We are therefore not having a total of 770 landslides. In addition, Broeckx et al (2018) and Nibigira (2019) contain also deep-seated landslides. Furthermore may entries in this dataset include mass movements associated with large gully features and with river bank erosion. These processes, in addition to not being landslides, are also strongly associated with human activities in the region (Dewitte et al., 2021; Kubwimana et al., 2021). Lastly, we shall also keep mind that the dataset of Broeckx et al (2018) is spatially biased towards the city of Bujumbura where image availability and density are higher than in other parts of the country, especially at the time when the inventory was compiled (see Depicker et al., 2021a; Figure 5). The dataset of Nibigira is focussed on only two regions of Burundi, which also leads to a spatial bias in the analysis.

Research in the region has shown that landscape rejuvenation due to the presence of migrating knickpoints associated with the rifting faults plays a major role in the distribution of the landslides. This is demonstrated at regional (Depicker et al., 2021a; 2024) and local levels (Kubwimana et al., 2021) for different types of landslides processes, whether form purely natural origin or from conditions associated with human activities (e.g. deforestation). Such influences of the rift is not even mentioned in the manuscript.

For the temporal analysis associated with landslides, reference to existing assessments with respect to landslide mobilisation rates (Depicker et al., 2021b, 2024) and rainfall thresholds (Monsieurs et al. 2019a, 2019b) would be welcome.

The climate change analysis was carried out with a rather straightforward analysis. I am surprised that such analysis is not carried out with reference to the state of the art (for example Souverijns et al., 2016) and the fact that conclusions on the issues of climate change are

difficult to draw in the region due to the absence of relevant data; which often leads to conflicting perspectives (IPCC, 2021).

- **Target audiences**

As stated in lines 40-42: "The results of the assessment were aimed at decision makers, civil protection authorities and other stakeholders at national and sub-national levels to support planning, decision-making and prioritisation of Disaster Risk Management (DRM) investments and activities.", the motivation of this research is to provide assessments for a specific audience. In that context, I find it a bit strange that the involvement of researchers from local institution is not considered. Usually, as expert, we are usually pleased to be invited to take part to a research were we can bring our own expertise. That would also have been a better strategy to try to overcome this gap between science and policy where the stakeholders are usually barely listened (Gill et al., 2021). Local scientists are certainly better at making stakeholders aware of the problems of natural hazards.

- **Introduction**

The state of the art is missed, especially on multi-hazard risk assessment. It is thereover difficult to position the research beyond a simple case study in the literature. Furthermore, the introduction brings quite a substantial amount of methodological information. Overall, the introduction reads more like a technical report that an research paper.

Note that the introduction points to cascading and compounding hazards, issues that are barely addressed in the subsequent analysis.

- **Study area**

The description of the study area remains very basic. One could have expected for example that reference to the rifting context is mentioned. Rifting is associated with the presence of faults and differences in relief, which has implication for earthquake hazard, local climate, and, as said earlier, landslide hazard.

The DRR context remains very general, relying on EM-DAT disaster data, a database that is known to come with some caveats. No reference is made to the local knowledge about the hazard (see earlier comments and the non-exhaustive list of references provided at the end of my comments).

Figure 1. What data sources are being used for the map? Where does the classification map of the landform come from? What is the used of such an information about the forms?

- **Methodology**

In addition to what I mention above about the landslide assessment, a lot of methodological steps and choices are not clearly justified and the descriptions of the methods are overall too superficial, preventing any reproducibility. Reference to the literature is very limited, hence

leaving the readers with questions about the relevance and reliability about the methodological aspects.

For example, for the climate drivers, no justification is made with respect to the use of the climate models to extract climate information. Why these models? Why not other products? There is quite a lot of literature on climate product comparison (even for the region – e.g. Camberlin et al., 2019; Nkunzimana et al., 2020); something of importance especially with regard to the specific climate variables that are to be used for the hazard assessments.

For the pluvial flooding, the collection of the information is not explained. The authors says that they have an inventory of 64 events. How? For exemple, Monsieurs et al '2018) and Nsabimana et al. (2023) did similar work in the region bringing enough information for making sure that the method can be reproduced.

For the flooding, another example is with this statement "geomorphological analyses based on digital terrain model were carried out in those flood-prone areas that were insufficiently covered by the historical flood information. " that is made without brining extra information. This is much too vague.

For the vulnerability for the landslides, the authors invoke a lack of literature. Such work in an neighbour environment similar to that of Burundi could be useful (Sekajugo et al., 2024)

- **Discussion**

This part should be a key aspect of the research. However a proper discussion is missed and there is nothing that is said with respect to the existing assessments. What is the added value of this research? What is its use? Where are the caveats?

The authors put an emphasis on climate change, which is of course a valid point. However, the main concerns about the natural hazards in this region are the exposure of the population and the weakness of the management (see for example Raju et al., 2022 that discuss such aspects in general). In addition, the impacts of human activities in the incidence of natural hazards such as landslides and floods (for example Depicker et al., 2021a, 2024) and the implication it has for the risk (Depicker et al., 2021b) are clear. These are points that for such as work would need to also be discussed, especially for a research that it aimed at targeting stakeholders. Overall, in that sense, the contextualized aspect of this research is weak I believe.

To summarize, the authors propose a research that aims to tackle a lot of issues on different hazards, their vulnerability, and climate change related aspects. This is a very ambitious work. However, lack of (i) methodological justification, (ii) use of local knowledge, (iii) discussion with respect to previous work, and (iv) absence of state of the art literature are factors that weakens the quality of this work. In addition, the research shows a lack a connection with its assume target audience. Per say, that is an point that one could understand, especially with respect to the constraint of going in the field (COVID restriction) and connecting with the local experts and institutions. Nevertheless, one would then have assumed a more elaborate discussion on those aspects.

I hope that my comments will be helpful.

Olivier Dewitte

References

Broeckx, J., Vanmaercke, M., Duchateau, R., & Poesen, J. (2018). A data-based landslide susceptibility map of Africa. Earth-Science Reviews, 185, 102-121.

Camberlin, P., Barraud, G., Bigot, S., Dewitte, O., Makanzu Imwangana, F., Maki Mateso, J. C., ... & Samba, G. (2019). Evaluation of remotely sensed rainfall products over Central Africa. Quarterly Journal of the Royal Meteorological Society, 145(722), 2115-2138.

Deijns, A. A., Dewitte, O., Thiery, W., d'Oreye, N., Malet, J. P., & Kervyn, F. (2022). Timing landslide and flash flood events from SAR satellite: a regionally applicable methodology illustrated in African cloud-covered tropical environments. Natural Hazards and Earth System Sciences, 22(11), 3679-3700.

Deijns, A. A., Michéa, D., Déprez, A., Malet, J. P., Kervyn, F., Thiery, W., & Dewitte, O. (2024). A semi-supervised multi-temporal landslide and flash flood event detection methodology for unexplored regions using massive satellite image time series. ISPRS Journal of Photogrammetry and Remote Sensing, 215, 400-418.

Delvaux, D., Mulumba, J. L., Sebagenzi, M. N. S., Bondo, S. F., Kervyn, F., & Havenith, H. B. (2017). Seismic hazard assessment of the Kivu rift segment based on a new seismotectonic zonation model (western branch, East African Rift system). Journal of African Earth Sciences, 134, 831-855.

Depicker, A., Jacobs, L., Delvaux, D., Havenith, H. B., Mateso, J. C. M., Govers, G., & Dewitte, O. (2020). The added value of a regional landslide susceptibility assessment: The western branch of the East African Rift. Geomorphology, 353, 106886.

Depicker, A., Govers, G., Jacobs, L., Campforts, B., Uwihirwe, J., & Dewitte, O. (2021a). Interactions between deforestation, landscape rejuvenation, and shallow landslides in the North Tanganyika–Kivu rift region, Africa. Earth Surface Dynamics, 9(3), 445-462.

Depicker, A., Jacobs, L., Mboga, N., Smets, B., Van Rompaey, A., Lennert, M., ... & Govers, G. (2021b). Historical dynamics of landslide risk from population and forest-cover changes in the Kivu Rift. Nature sustainability, 4(11), 965-974.

Depicker, A., Govers, G., Jacobs, L., Vanmaercke, M., Uwihirwe, J., Campforts, B., ... & Dewitte, O. (2024). Mobilization rates of landslides in a changing tropical environment: 60-year record over a large region of the East African Rift. Geomorphology, 454, 109156.

Dewitte, O., Dille, A., Depicker, A., Kubwimana, D., Maki Mateso, J. C., Mugaruka Bibentyo, T., ... & Monsieurs, E. (2021). Constraining landslide timing in a data-scarce context: from recent to very old processes in the tropical environment of the North Tanganyika-Kivu Rift region. Landslides, 18(1), 161-177.

Gill, J. C., Taylor, F. E., Duncan, M. J., Mohadjer, S., Budimir, M., Mdala, H., and Bukachi, V.: Invited perspectives: Building sustainable and resilient communities – recommended actions for natural hazard scientists, Nat. Hazards Earth Syst. Sci., 21, 187–202, https://doi.org/10.5194/nhess-21-187-2021, 2021.

Kubwimana, D., Ait Brahim, L., Nkurunziza, P., Dille, A., Depicker, A., Nahimana, L., ... & Dewitte, O. (2021). Characteristics and distribution of landslides in the populated hillslopes of Bujumbura, Burundi. Geosciences, 11(6), 259.

IPCC, 2021: Climate Change 2021: The Physical Science Basis. Contribution of Working Group I to the Sixth Assessment Report of the Intergovernmental Panel on Climate Change[Masson-Delmotte, V., P. Zhai, A. Pirani, S.L. Connors, C. Péan, S. Berger, N. Caud, Y. Chen, L. Goldfarb, M.I. Gomis, M. Huang, K. Leitzell, E. Lonnoy, J.B.R. Matthews, T.K. Maycock, T. Waterfield, O. Yelekçi, R. Yu, and B. Zhou (eds.)]. Cambridge University Press, Cambridge, United Kingdom and New York, NY, USA, In press, doi:10.1017/9781009157896.

Monsieurs, E., Jacobs, L., Michellier, C., Basimike Tchangaboba, J., Ganza, G. B., Kervyn, F., ... & Dewitte, O. (2018). Landslide inventory for hazard assessment in a data-poor context: a regional-scale approach in a tropical African environment. Landslides, 15, 2195-2209.

Monsieurs, E., Dewitte, O., & Demoulin, A. (2019a). A susceptibility-based rainfall threshold approach for landslide occurrence. Natural Hazards and Earth System Sciences, 19(4), 775-789.

Monsieurs, E., Dewitte, O., Depicker, A., & Demoulin, A. (2019b). Towards a transferable antecedent rainfall—susceptibility threshold approach for landsliding. Water, 11(11), 2202.

Nibigira, L., Havenith, H. B., Archambeau, P., & Dewals, B. (2018). Formation, breaching and flood consequences of a landslide dam near Bujumbura, Burundi. Natural Hazards and Earth System Sciences, 18(7), 1867-1890.

Nkunzimana, A., Bi, S., Alriah, M. A. A., Zhi, T., & Kur, N. A. D. (2020). Comparative analysis of the performance of satellite-based rainfall products over various topographical unities in Central East Africa: case of Burundi. Earth and Space Science, 7(5), e2019EA000834.

Nsabimana, J., Henry, S., Ndayisenga, A., Kubwimana, D., Dewitte, O., Kervyn, F., & Michellier, C. (2023). Geo-hydrological hazard impacts, vulnerability and perception in Bujumbura (Burundi): a high-resolution field-based assessment in a sprawling city. Land, 12(10), 1876.

Raju, E., Boyd, E., & Otto, F. (2022). Stop blaming the climate for disasters. Communications Earth & Environment, 3(1), 1.

Sekajugo, J., Kagoro-Rugunda, G., Mutyebere, R., Kabaseke, C., Mubiru, D., Kanyiginya, V., ... & Kervyn, M. (2024). Exposure and physical vulnerability to geo-hydrological hazards in rural environments: A field-based assessment in East Africa. International Journal of Disaster Risk Reduction, 102, 104282.

Souverijns, N., Thiery, W., Demuzere, M., & Van Lipzig, N. P. (2016). Drivers of future changes in East African precipitation. Environmental research letters, 11(11), 114011.

---

## Referee Comment (RC2)

**Dear Jess Delves, Kathrin Renner, Piero Campalani, Jesica Piñón, Stefan Schneiderbauer, Stefan Steger, Mateo Moreno, Maria Belen Benito Oterino, Eduardo Perez, and Massimiliano Pittore**

**Dear editor, Robert Sakic Trogrlic,**

I believe some of the content in this paper is relevant for publication in NHESS, and certain findings may have implications for disaster risk management. However, due to the manuscript's disorganized structure and specific imprecisions, substantial revisions are necessary before moving forward. I found it challenging to identify any clear innovations or methodological advancements from the manuscript. Simply applying existing state-of-the-art (or at least state-of-practice) tools to a different problem or case study does not inherently provide novelty. Nevertheless, deriving new insights—even from straightforward applications— can be valuable and appropriate. Unfortunately, the authors do not present a compelling case for any new insights gained through their application of established tools to a new case study.

Despite significant shortcomings, I opted not to reject the manuscript at first, as I recognized the hard work that went into assembling these ideas and the potential for application and replication of the study. My recommendation is that major revisions be undertaken prior to publication, as the manuscript in its current form does not meet the quality standards expected.

I have provided the authors with a first review to assist in improving the quality of their manuscript. I am confident that with substantial revisions addressing the feedback I outline below, the quality of the paper could be enhanced to meet NHESS standards. I advise that all authors contribute to a comprehensive revision of the manuscript and reach an agreement on its content before resubmission. The responsibility is not solely that of the main author. I believe the manuscript will improve significantly with the notable experience and careful proofreading of senior scientists.

However, given the nature of these comments, the decision ultimately rests with the Editor as to whether the manuscript can indeed undergo major revisions or if it should be rejected with the possibility of a new submission.

**A. MAJOR COMMENTS**

1. **Style:**
   The paper appears to be more of a project report rather than a scientific manuscript, evident from its general structure and specific phrasing (e.g., lines 36, 43, 352, etc.). The scope of the study seems directed towards a specific audience, distinct from the scientific community, which is acceptable. However, the basics of the various methodologies and assumptions should be clearly stated, even if only in a supplementary section. Below are some related suggestions.

2. **Originality:**
   I have concerns that this paper may have been submitted elsewhere (referencing https://papers.ssrn.com/sol3/papers.cfm?abstract_id=4882187) or that it may be part of an existing project report or deliverable that is currently available or will be published in the future, raising potential originality conflicts. If this is the case, this should be noted in the manuscript.

3. **Structure**

   **3.1.** Considering the final results the paper intends to convey, the structure of the submitted manuscript requires revision. Numerous sub-sections lack sufficient detail (e.g., the approaches followed for hazard-specific vulnerabilities) and surface late in the document. It would be beneficial to consolidate sections by creating "chunks" for each hazard-related physical risk. This means having risk as a header, with hazard and physical vulnerabilities (the adopted methods) presented together. Such an organization would harmonize text that currently mixes these components (e.g., lines 173-175 & 235-240) in the "Hazard" section, while exposure and social vulnerability could be presented separately.

   **3.2.** The current arrangement of having discussion as Section 9 (after Section 8: Results) and then Section 10 on 'Climate Change' is not appealing. I advise revising this structure.

   **3.3.** The SEVA and Climate Change analyses were not integrated into the final results and therefore feel distracting. I suggest either completely removing them or presenting them as an Appendix, treating them as auxiliary analyses rather than part of the main text. The same applies to the details for the flood model for the five catchments (Fig. 4), as this is also distracting.

4. **Material and methods**

   **4.1.** My understanding is that the left subplot in Fig. 7 derives entirely from Worldpop (the full citation is missing). However, it is unclear which spatial disaggregation technique was used to derive building counts on the right. No equations or methods were provided. The existing information does not sufficiently connect with the risk assessment. How do these counts compare to existing datasets such as IOM? What methodology (i.e., equations) was used to adjust these initial numbers? Why were existing datasets like the Global Human Settlement – GHSL not utilized, given they offer estimates on the number of buildings and even future populations at higher resolutions?

   **4.2.** My understanding is that population (from Worldpop) was used solely to derive building counts, with no risk-related metrics developed (e.g., casualties, human displacement); only monetary risk metrics were assessed. This should be clearly stated from the outset.

   **4.3.** Line 292 states that GEM taxonomy Version 2.0 is focused entirely on seismic vulnerability applications, including various attributes. This statement is inaccurate. For Fig. 8, only material and number of stories were included in the ten baseline classes. The authors claim that GEM and DTM building types were merged, but the basis for this is unclear. How was this validation conducted? If a test was performed, why not present it as supplementary information?

   **4.4.** The manuscript does not provide the spatial distribution of repair costs for residential buildings and crop values, nor does it include this information in a figure or table, or detail its derivation. This is crucial considering that a key result is the assessment of the Annual Average Loss (AAL). One option might be to include another subplot in Fig. 7 showcasing these repair cost values.

   **4.5.** Regarding damage functions for crops, the only mention is in line 115: "for the impact of fluvial flooding on agricultural areas and for landslides, a simpler binary fragility model was used due to a lack of consistent alternatives." This aspect requires more detail. What thresholds were used to determine loss? Was a hazard intensity or distance-based metric assumed for vulnerability to landslides?

**4.6.** Vulnerability of buildings to landslides: In line 354, what exactly does "given distance" from the location assigned to a landslide event refer to?

**4.7.** SEVA.

    **4.7.1.** Line 342: notes that coping capacity may be one of the highlights and novelties of the manuscript. However, little detail on the methodology and its impact on final results is provided. This should be commented on in the conclusions.

    **4.7.2.** Line 361, which proxies?

    **4.7.3.** terms "indicators, components, categories, dimensions" is not well separated, and sometimes used interchangeably.

**5. Main results:**

**5.1.** In the multi-hazard map shown in Fig. 12, why was the 85th percentile selected? This assumption has not been discussed or justified. Notably, the same percentile for different (and very different) return periods has been combined. How can this be technically justified?

**5.2.** The final results presented in Fig. 13 come from aggregating the individual results in Table 1. My understanding is that 5 out of the 6 AAL columns were derived from assessing the risk to residential buildings, while only one column reflects the potential losses of crops due to river floods. This indicates that the vulnerability of crops was not assessed for torrential rains, winds, landslides, or earthquakes. This separation raises concerns.

**5.3.** The final result in Fig. 13 was obtained by dividing the physical risk by the area of the colline. This appears to be an arbitrary assumption; various results would arise if the AAL risk were divided by average gross income, total population, or any other aggregated parameter. This approach needs revision, commentary, and stronger justification.

**5.4.** Although I previously suggested moving the impact of climate change to an Appendix, I am concerned about lines 470-472, where the authors mention that Fig. 15 is derived from averaging three "future projections" (global climatic models) and "three return periods." Is this map an average of three return periods and three climatic models? The authors should provide a justification or methodology for this assumption. How should these results be interpreted, and are they genuinely useful for practical applications?

**6. References**

**6.1.** Official citations for datasets and other relevant sources are missing (e.g., Worldpop, IPCC).

**6.2.** Given the difficulty in assessing impacts, open sources, such as the Climate Impact Explorer (https://climate-impact-explorer.climateanalytics.org/impacts/) and the ISIMIP repository (https://data.isimip.org/), could have offered valuable initial insights. Additionally, as pointed out by Reviewer 1, several existing datasets have been completely neglected.

**6.3.** Several studies on flood vulnerability and risk in the African context were conducted under the CLUVA project. I recommend that the authors include and comment on these works (e.g., Jalayer et al., 2014, 2016).

**6.4.** The entire section "5.4 Earthquakes" lacks bibliographic references. This omission implies that the authors claim all content therein as their own, which is likely not true. Also, in line 254, the authors mention five hotspot localities without further elaboration. If this is not part of the final results, why is it included?

**6.5.** The references of Paul et al., (2022); and Paul and Silva, (2025) are missing. are missing. These studies contain valuable references that should be contrasted with the authors' findings.
There has been no thorough bibliographic revision concerning multi-hazard risk. I fully agree with Reviewer 1 on this point.

7. **Data availability**

Access to geodata products is not only pertinent to reviewers; potential future readers may seek access to replicate the methodology using the developed tools in other study areas. This is crucial for adhering to and upholding the FAIR principles (Findable, Accessible, Interoperable, and Reusable) (Wilkinson et al., 2016).

B. **Minor comments:**

1. Figures. In general, all figures with maps should include clear copyright statements.
    **1.1.** Fig. 2: should be improved. It is not self-explanatory. I advise to include references for the adopted models.
    **1.2.** Fig. 10: The legend was never mentioned in the text. Also, it is unclear why this damage function is included when the adopted function for earthquakes is not shown. As this is not an original development, I recommend removing this figure.
    **1.3.** Fig. 7: I advise to use a continuous colour bar, and not discrete intervals.
    **1.4.** Fig. 12: delete the title. The figure's caption is enough. I advise to update the figure showing the two lower subplots and the table bigger.
    **1.5.** References style: The footnote 1 (line 288) is not a correct citation. Please note that in that website, the citation suggested is mentioned.
2. Please, add every individual author's contribution, as required by NHESS.

C. **Editorial comments:**

1. A more thorough review of the English writing is recommended, focusing not only on grammar but also on the presentation of ideas. I am particularly surprised that this was not noted by the senior co-authors of the manuscript, who should have provided careful supervision and strict approval before resubmission.
2. The use of several adjectives, such as "accurately" (line 456), "precise" (line 449), and "effectively" (line 457), is inappropriate given the uncertainties associated with global climatic models. I advise using more straightforward language.
3. Please ensure uniform font style throughout the document. The text currently displays mixed font styles that do not comply with the NHESS template (e.g., line 212).
4. In line 299, the phrase "finally..." implies that the presentation of the figure in pie charts is part of the methodology. This is misleading, as it represents merely a selected format of presentation.
5. There are several disconnected sentences that appear as if different texts were mixed together without harmonization by a careful reader.
6. Similarly to Reviewer 1, I am surprised that no local authors were invited to contribute to this study. I recommend that the authors consult the EGU statement on Scientific Neocolonialism.

*Best regards.*

**References**

Jalayer, F., Risi, R., Paola, F., Giugni, M., Manfredi, G., Gasparini, P., Topa, M., Yonas, N., Yeshitela, K., Nebebe, A., Cavan, G., and Lindley, S.: Probabilistic GIS-based method for delineation of urban flooding risk hotspots, Natural Hazards: Journal of the International Society for the Prevention and Mitigation of Natural Hazards, 73, 975–1001, https://doi.org/10.1007/s11069-014-1119-2, 2014.

Jalayer, F., Carozza, S., De Risi, R., Manfredi, G., and Mbuya, E.: Performance-based flood safety-checking for non-engineered masonry structures, Engineering Structures, 106, 109–123, https://doi.org/10.1016/j.engstruct.2015.10.007, 2016.

Paul, N. and Silva, V.: Probabilistic seismic risk assessment of Africa, International Journal of Disaster Risk Reduction, 119, 105303, https://doi.org/10.1016/j.ijdrr.2025.105303, 2025.

Paul, N., Silva, V., and Amo-Oduro, D.: Development of a uniform exposure model for the African continent for use in disaster risk assessment, International Journal of Disaster Risk Reduction, 71, 102823, https://doi.org/10.1016/j.ijdrr.2022.102823, 2022.

Wilkinson, M. D., Dumontier, M., Aalbersberg, Ij. J., Appleton, G., Axton, M., Baak, A., Blomberg, N., Boiten, J.-W., da Silva Santos, L. B., Bourne, P. E., Bouwman, J., Brookes, A. J., Clark, T., Crosas, M., Dillo, I., Dumon, O., Edmunds, S., Evelo, C. T., Finkers, R., Gonzalez-Beltran, A., Gray, A. J. G., Groth, P., Goble, C., Grethe, J. S., Heringa, J., 't Hoen, P. A. C., Hooft, R., Kuhn, T., Kok, R., Kok, J., Lusher, S. J., Martone, M. E., Mons, A., Packer, A. L., Persson, B., Rocca-Serra, P., Roos, M., van Schaik, R., Sansone, S.-A., Schultes, E., Sengstag, T., Slater, T., Strawn, G., Swertz, M. A., Thompson, M., van der Lei, J., van Mulligen, E., Velterop, J., Waagmeester, A., Wittenburg, P., Wolstencroft, K., Zhao, J., and Mons, B.: The FAIR Guiding Principles for scientific data management and stewardship, Scientific Data, 3, 160018, https://doi.org/10.1038/sdata.2016.18, 2016.